# Efficient Learning of Generative Models via Finite-Difference Score Matching

**Tianyu Pang**[*1], **Kun Xu**[*1], **Chongxuan Li**[1], **Yang Song**[2], **Stefano Ermon**[2], **Jun Zhu**[†1]

[1]Dept. of Comp. Sci. & Tech., Institute for AI, BNRist Center,
Tsinghua-Bosch Joint ML Center, THBI Lab, Tsinghua University
[2]Department of Computer Science, Stanford University
pty17@mails.tsinghua.edu.cn, {kunxu.thu, chongxuanli1991}@gmail.com
{yangsong, ermon}@cs.stanford.edu, dcszj@mail.tsinghua.edu.cn

## Abstract

Several machine learning applications involve the optimization of higher-order derivatives (e.g., gradients of gradients) during training, which can be expensive with respect to memory and computation even with automatic differentiation. As a typical example in generative modeling, score matching (SM) involves the optimization of the trace of a Hessian. To improve computing efficiency, we rewrite the SM objective and its variants in terms of directional derivatives, and present a generic strategy to efficiently approximate any-order directional derivative with finite difference (FD). Our approximation only involves function evaluations, which can be executed in parallel, and no gradient computations. Thus, it reduces the total computational cost while also improving numerical stability. We provide two instantiations by reformulating variants of SM objectives into the FD forms. Empirically, we demonstrate that our methods produce results comparable to the gradient-based counterparts while being much more computationally efficient.

## 1 Introduction

Deep generative models have achieved impressive progress on learning data distributions, with either an explicit density function [24, 26, 46, 48] or an implicit generative process [1, 10, 75]. Among explicit models, energy-based models (EBMs) [34, 63] define the probability density as $p_\theta(x) = \widetilde{p}_\theta(x)/Z_\theta$, where $\widetilde{p}_\theta(x)$ denotes the unnormalized probability and $Z_\theta = \int \widetilde{p}_\theta(x)dx$ is the partition function. EBMs allow more flexible architectures [9, 11] with simpler compositionality [15, 41] compared to other explicit generative models [46, 16], and have better stability and mode coverage in training [30, 31, 71] compared to implicit generative models [10]. Although EBMs are appealing, training them with maximum likelihood estimate (MLE), i.e., minimizing the KL divergence between data and model distributions, is challenging because of the intractable partition function [20].

Score matching (SM) [21] is an alternative objective that circumvents the intractable partition function by training unnormalized models with the Fisher divergence [23], which depends on the Hessian trace and (Stein) score function [37] of the log-density function. SM eliminates the dependence of the log-likelihood on $Z_\theta$ by taking derivatives w.r.t. $x$, using the fact that $\nabla_x \log p_\theta(x) = \nabla_x \log \widetilde{p}_\theta(x)$. Different variants of SM have been proposed, including approximate back-propagation [25], curvature propagation [39], denoising score matching (DSM) [65], a bi-level formulation for latent variable models [3] and nonparametric estimators [35, 55, 59, 62, 74], but they may suffer from high computational cost, biased parameter estimation, large variance, or complex implementations. Sliced score matching (SSM) [58] alleviates these problems by providing a scalable and unbiased estimator

---

[*]Equal contribution. [†] Corresponding author.

with a simple implementation. However, most of these score matching methods optimize (high-order) derivatives of the density function, e.g., the gradient of a Hessian trace w.r.t. parameters, which are several times more computationally expensive compared to a typical end-to-end propagation, even when using reverse-mode automatic differentiation [13, 47]. These extra computations need to be performed in sequential order and cannot be easily accelerated by parallel computing (as discussed in Appendix B.1). Besides, the induced repetitive usage of the same intermediate results could magnify the stochastic variance and lead to numerical instability [60].

To improve efficiency and stability, we first observe that existing scalable SM objectives (e.g., DSM and SSM) can be rewritten in terms of (second-order) directional derivatives. We then propose a generic finite-difference (FD) decomposition for any-order directional derivative in Sec. 3, and show an application to SM methods in Sec. 4, eliminating the need for optimizing on higher-order gradients. Specifically, our FD approach only requires independent (unnormalized) likelihood function evaluations, which can be efficiently and synchronously executed in parallel with a simple implementation (detailed in

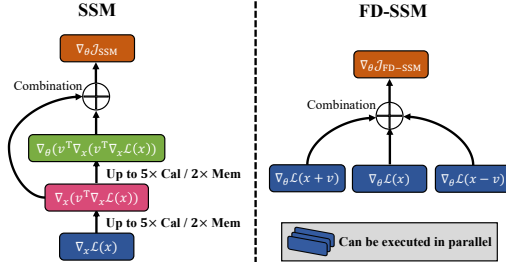

Figure 1: Computing graphs of each update step. Detailed in Sec. 2.2 (SSM) and Sec. 4 (FD-SSM).

Sec. 3.3). This approach reduces the computational complexity of any $T$-th order directional derivative to $\mathcal{O}(T)$, and improves numerical stability because it involves a shallower computational graph. As we exemplify in Fig. 1, the FD reformulations decompose the inherently sequential high-order gradient computations in SSM (left panel) into simpler, independent routines (right panel). Mathematically, in Sec. 5 we show that even under stochastic optimization [51], our new FD objectives are asymptotically consistent with their gradient-based counterparts under mild conditions. When the generative models are unnormalized, the intractable partition function can be eliminated by the linear combinations of log-density in the FD-form objectives. In experiments, we demonstrate the speed-up ratios of our FD reformulations with more than $2.5\times$ for SSM and $1.5\times$ for DSM on different generative models and datasets, as well as the comparable performance of the learned models.

## 2 Background

Explicit generative modeling aims to model the true data distribution $p_{\text{data}}(x)$ with a parametric model $p_\theta(x)$, where $x \in \mathbb{R}^d$. The learning process usually minimizes some divergence between $p_\theta(x)$ and the (empirical) data distribution (e.g., KL-divergence minimization leads to MLE). In particular, the unnormalized generative models such as the energy-based ones [34] model the distribution as $p_\theta(x) = \widetilde{p}_\theta(x)/Z_\theta$, where $\widetilde{p}_\theta(x)$ is the unnormalized probability and $Z_\theta = \int \widetilde{p}_\theta(x)dx$ is the partition function. Computing the integral in $Z_\theta$ is usually intractable especially for high-dimensional data, which makes it difficult to directly learn unnormalized models with MLE [9, 29].

### 2.1 Score matching methods

As an alternative to KL divergence, score matching (SM) [21] minimizes the Fisher divergence between $p_\theta(x)$ and $p_{\text{data}}(x)$, which is equivalent to

$$\mathcal{J}_{\text{SM}}(\theta) = \mathbb{E}_{p_{\text{data}}(x)}\left[\text{tr}(\nabla_x^2 \log p_\theta(x)) + \frac{1}{2}\|\nabla_x \log p_\theta(x)\|_2^2\right] \tag{1}$$

up to a constant and $\text{tr}(\cdot)$ is the matrix trace. Note that the derivatives w.r.t. $x$ eliminate the dependence on the partition function, i.e., $\nabla_x \log p_\theta(x) = \nabla_x \log \widetilde{p}_\theta(x)$, making the objective function tractable. However, the calculation of the trace of Hessian matrix is expensive, requiring the number of back-propagations proportional to the data dimension [39]. To circumvent this computational difficulty, two scalable variants of SM have been developed, to which we will apply our methods.

**Denoising score matching (DSM).** Vincent [65] circumvents the Hessian trace by perturbing $x$ with a noise distribution $p_\sigma(\widetilde{x}|x)$ and then estimating the score of the perturbed data distribution

$p_\sigma(\widetilde{x}) = \int p_\sigma(\widetilde{x}|x)p_{\text{data}}(x)dx$. When using Gaussian noise, we obtain the DSM objective as

$$\mathcal{J}_{\text{DSM}}(\theta) = \frac{1}{d}\mathbb{E}_{p_{\text{data}}(x)}\mathbb{E}_{p_\sigma(\widetilde{x}|x)}\left[\left\|\nabla_{\widetilde{x}}\log p_\theta(\widetilde{x}) + \frac{\widetilde{x}-x}{\sigma^2}\right\|_2^2\right], \tag{2}$$

The model obtained by DSM only matches the true data distribution when the noise scale $\sigma$ is small enough. However, when $\sigma \to 0$, the variance of DSM could be large or even tend to infinity [66], requiring grid search or heuristics for choosing $\sigma$ [53].

**Sliced score matching (SSM).** Song et al. [58] use random projections to avoid explicitly calculating the Hessian trace, so that the training objective only involves Hessian-vector products as follows:

$$\mathcal{J}_{\text{SSM}}(\theta) = \frac{1}{C_v}\mathbb{E}_{p_{\text{data}}(x)}\mathbb{E}_{p_v(v)}\left[v^\top\nabla_x^2\log p_\theta(x)v + \frac{1}{2}\left(v^\top\nabla_x\log p_\theta(x)\right)^2\right], \tag{3}$$

where $v \sim p_v(v)$ is the random direction, $\mathbb{E}_{p_v(v)}[vv^\top] \succ 0$ and $C_v = \mathbb{E}_{p_v(v)}[\|v\|_2^2]$ is a constant w.r.t. $\theta$. We divide the SSM loss by $C_v$ to exclude the dependence on the scale of the projection distribution $p_v(v)$. Here $p_{\text{data}}(x)$ and $p_v(v)$ are independent. Unlike DSM, the model obtained by SSM can match the original unperturbed data distribution, but requires more expensive, high-order derivatives.

## 2.2 Computational cost of gradient-based SM methods

Although SM methods can bypass the intractable partition function $Z_\theta$, they have to optimize an objective function involving higher-order derivatives of the log-likelihood density. Even if reverse mode automatic differentiation is used [47], existing SM methods like DSM and SSM can be computationally expensive during training when calculating the Hessian-vector products.

**Complexity of the Hessian-vector products.** Let $\mathcal{L}$ be any loss function, and let $\text{Cal}(\nabla\mathcal{L})$ and $\text{Mem}(\nabla\mathcal{L})$ denote the time and memory required to compute $\nabla\mathcal{L}$, respectively. Then if the reverse mode of automatic differentiation is used, the Hessian-vector product can be computed with up to five times more time and two times more memory compared to $\nabla\mathcal{L}$, i.e., $5\times \text{Cal}(\nabla\mathcal{L})$ time and $2\times \text{Mem}(\nabla\mathcal{L})$ memory [12, 13]. When we instantiate $\mathcal{L} = \log p_\theta(x)$, we can derive that the computations of optimizing DSM and SSM are separately dominated by the sequential operations of $\nabla_\theta(\|\nabla_x\mathcal{L}\|)$ and $\nabla_\theta(v^\top\nabla_x(v^\top\nabla_x\mathcal{L}))$, as illustrated in Fig. 1 for SSM. The operations of $\nabla_\theta$ and $\nabla_x$ require comparable computing resources, so we can conclude that compared to directly optimizing the log-likelihood, DSM requires up to $5\times$ computing time and $2\times$ memory, while SSM requires up to $25\times$ computing time and $4\times$ memory [12]. For higher-order derivatives, we empirically observe that the computing time and memory usage grow exponentially w.r.t. the order of derivatives, i.e., the times of executing the operator $v^\top\nabla$, as detailed in Sec. 3.3.

## 3 Approximating directional derivatives via finite difference

In this section, we first rewrite the most expensive terms in the SM objectives in terms of directional derivatives, then we provide generic and efficient formulas to approximate any $T$-th order directional derivative using finite difference (FD). The proposed FD approximations decompose the sequential and dependent computations of high-order derivatives into independent and parallelizable computing routines, reducing the computational complexity to $\mathcal{O}(T)$ and improving numerical stability.

### 3.1 Rewriting SM objectives in directional derivatives

Note that the objectives of SM, DSM, and SSM described in Sec. 2.1 can all be abstracted in terms of $v^\top\nabla_x\mathcal{L}_\theta(x)$ and $v^\top\nabla_x^2\mathcal{L}_\theta(x)v$. Specifically, as to SM or DSM, $v$ is the basis vector $e_i$ along the $i$-th coordinate to constitute the squared norm term $\|\nabla_x\mathcal{L}_\theta(x)\|_2^2 = \sum_{i=1}^d(e_i^\top\nabla_x\mathcal{L}_\theta(x))^2$ or the Hessian trace term $\text{tr}(\nabla_x^2\mathcal{L}_\theta(x)) = \sum_{i=1}^d e_i^\top\nabla_x^2\mathcal{L}_\theta(x)e_i$. As to SSM, $v$ denotes the random direction.

We regard the gradient operator $\nabla_x$ as a $d$-dimensional vector $\nabla_x = (\frac{\partial}{\partial x_1}, \cdots, \frac{\partial}{\partial x_d})$, and $v^\top\nabla_x$ is an operator that first executes $\nabla_x$ and then projects onto the vector $v$. For notation simplicity, we denote $\|v\|_2 = \epsilon$ and rewrite the above terms as (higher-order) directional derivatives as follows:

$$v^\top\nabla_x = \epsilon\frac{\partial}{\partial v}; \quad v^\top\nabla_x\mathcal{L}_\theta(x) = \epsilon\frac{\partial}{\partial v}\mathcal{L}_\theta(x); \quad v^\top\nabla_x^2\mathcal{L}_\theta(x)v = (v^\top\nabla_x)^2\mathcal{L}_\theta(x) = \epsilon^2\frac{\partial^2}{\partial v^2}\mathcal{L}_\theta(x). \tag{4}$$

Here $\frac{\partial}{\partial v}$ is the directional derivative along $v$, and $\left(v^\top\nabla_x\right)^2$ means executing $v^\top\nabla_x$ twice.

### 3.2   FD decomposition for directional derivatives

We propose to adopt the FD approach, a popular tool in numerical analysis to approximate differential operations [60], to efficiently estimate the terms in Eq. (4). Taking the first-order case as an example, the key idea is that we can approximate $\frac{\partial}{\partial v}\mathcal{L}_\theta(x) = \frac{1}{2\epsilon}(\mathcal{L}_\theta(x+v) - \mathcal{L}_\theta(x-v)) + o(\epsilon)$, where the right-hand side does not involve derivatives, just function evaluations. In FD, $\|v\|_2 = \epsilon$ is assumed to be a small value, but this does not affect the optimization of SM objectives. For instance, the SSM objective in Eq. (3) can be adaptively rescaled by $C_v$ (generally explained in Appendix B.2).

In general, to estimate the $T$-th order directional derivative of $\mathcal{L}_\theta$, which is assumed to be $T$ times differentiable, we first apply the multivariate Taylor's expansion with Peano's remainder [27] as

$$\mathcal{L}_\theta(x + \gamma v) = \sum_{t=0}^{T} \frac{\gamma^t}{t!} \left(v^\top \nabla_x\right)^t \mathcal{L}_\theta(x) + o(\epsilon^T) = \sum_{t=0}^{T} \gamma^t \left(\frac{\epsilon^t}{t!} \frac{\partial^t}{\partial v^t} \mathcal{L}_\theta(x)\right) + o(\epsilon^T), \quad (5)$$

where $\gamma \in \mathbb{R}$ is a certain coefficient. Then, we take a linear combination of the Taylor expansion in Eq. (5) for different values of $\gamma$ and eliminate derivative terms of order less than $T$. Formally, $T+1$ different $\gamma$s are sufficient to construct a valid FD approximation (all the proofs are in Appendix A).[1]

**Lemma 1.** *(Existence of $o(1)$ estimator) If $\mathcal{L}_\theta(x)$ is $T$-times-differentiable at $x$, then given any set of $T + 1$ different real values $\{\gamma_i\}_{i=1}^{T+1}$, there exist corresponding coefficients $\{\beta_i\}_{i=1}^{T+1}$, such that*

$$\frac{\partial^T}{\partial v^T}\mathcal{L}_\theta(x) = \frac{T!}{\epsilon^T} \sum_{i=1}^{T+1} \beta_i \mathcal{L}_\theta(x + \gamma_i v) + o(1). \quad (6)$$

Lemma 1 states that it is possible to approximate the $T$-th order directional derivative as to an $o(1)$ error with $T+1$ function evaluations. In fact, as long as $\mathcal{L}_\theta(x)$ is $(T+1)$-times-differentiable at $x$, we can construct a special kind of linear combination of $T+1$ function evaluations to reduce the approximation error to $o(\epsilon)$, as stated below:

**Theorem 1.** *(Construction of $o(\epsilon)$ estimator) If $\mathcal{L}_\theta(x)$ is $(T+1)$-times-differentiable at $x$, we let $K \in \mathbb{N}^+$ and $\{\alpha_k\}_{k=1}^{K}$ be any set of $K$ different positive numbers, then we have the FD decomposition*

$$\frac{\partial^T}{\partial v^T}\mathcal{L}_\theta(x) = o(\epsilon) + \begin{cases} \dfrac{T!}{2\epsilon^T} \displaystyle\sum_{k\in[K]} \beta_k \alpha_k^{-2} \left[\mathcal{L}_\theta(x+\alpha_k v) + \mathcal{L}_\theta(x-\alpha_k v) - 2\mathcal{L}_\theta(x)\right], \text{ when } T = 2K; \\[2ex] \dfrac{T!}{2\epsilon^T} \displaystyle\sum_{k\in[K]} \beta_k \alpha_k^{-1} \left[\mathcal{L}_\theta(x+\alpha_k v) - \mathcal{L}_\theta(x-\alpha_k v)\right], \text{ when } T = 2K - 1. \end{cases} \quad (7)$$

*The coefficients $\boldsymbol{\beta} \in \mathbb{R}^K$ is the solution of $V^\top \boldsymbol{\beta} = \boldsymbol{e}_K$, where $V \in \mathbb{R}^{K \times K}$ is the Vandermonde matrix induced by $\{\alpha_k^2\}_{k=1}^{K}$, i.e., $V_{ij} = \alpha_i^{2j-2}$, and $\boldsymbol{e}_K \in \mathbb{R}^K$ is the $K$-th basis vector.*

It is easy to generalize Theorem 1 to achieve approximation error $o(\epsilon^N)$ for any $N \geq 1$ with $T+N$ function evaluations, and we can show that the error rate $o(\epsilon)$ is optimal when evaluating $T+1$ functions. So far we have proposed generic formulas for the FD decomposition of any-order directional derivative. As to the application to SM objectives (detailed in Sec. 4), we can instantiate the decomposition in Theorem 1 with $K = 1$, $\alpha_1 = 1$, and solve for $\beta_1 = 1$, which leads to

$$\begin{cases} v^\top \nabla_x \mathcal{L}_\theta(x) = \epsilon \dfrac{\partial}{\partial v}\mathcal{L}_\theta(x) = \dfrac{1}{2}\mathcal{L}_\theta(x+v) - \dfrac{1}{2}\mathcal{L}_\theta(x-v) + o(\epsilon^2); \\[2ex] v^\top \nabla_x^2 \mathcal{L}_\theta(x)v = \epsilon^2 \dfrac{\partial^2}{\partial v^2}\mathcal{L}_\theta(x) = \mathcal{L}_\theta(x+v) + \mathcal{L}_\theta(x-v) - 2\mathcal{L}_\theta(x) + o(\epsilon^3). \end{cases} \quad (8)$$

In addition to generative modeling, the decomposition in Theorem 1 can potentially be used in other settings involving higher-order derivatives, e.g., extracting local patterns with high-order directional derivatives [72], training GANs with gradient penalty [40], or optimizing the Fisher information [5]. We leave these interesting explorations to future work.

**Remark.** When $\mathcal{L}_\theta(x)$ is modeled by a neural network, we can employ the average pooling layer and the non-linear activation of, e.g., Softplus [73] to have an infinitely differentiable model to meet the condition in Theorem 1. Note that Theorem 1 promises a *point-wise* approximation error $o(\epsilon)$. To validate the error rate *under expectation* for training objectives, we only need to assume that $p_{\text{data}}(x)$ and $\mathcal{L}_\theta(x)$ satisfy mild regularity conditions beyond the one in Theorem 1, which can be easily met in practice, as detailed in Appendix B.3. Conceptually, these mild regularity conditions enable us to substitute the Peano's remainders with Lagrange's ones. Moreover, this substitution results in a better approximation error of $\mathcal{O}(\epsilon^2)$ for our FD decomposition, while we still use $o(\epsilon)$ for convenience.

### 3.3 Computational efficiency of the FD decomposition

Theorem 1 provides a generic approach to approximate any $T$-th order directional derivative by decomposing the sequential and dependent order-by-order computations into independent function evaluations. This decomposition reduces the computational complexity to $\mathcal{O}(T)$, while the complexity of explicitly computing high-order derivatives usually grows exponentially w.r.t. $T$ [12], as we verify in Fig. 2. Furthermore, due to the mutual independence among the function terms $\mathcal{L}_\theta(x + \gamma_i v)$, they can be efficiently and synchronously executed in parallel via simple implementation (pseudo code is in Appendix C.1). Since this parallelization acts on the level of operations for each data point $x$, it is compatible with data or model parallelism to further accelerate the calculations.

To empirically demonstrate the computational efficiency of our FD decomposition, we report the computing time and memory usage in Fig. 2 for calculating the $T$-th order directional derivative, i.e., $\frac{\partial^T}{\partial v^T}$ or $(v^\top \nabla_x)^T$, either exactly or by the FD decomposition. The function $\mathcal{L}_\theta(x)$ is the log-density modeled by a deep EBM and trained on MNIST, while we use PyTorch [47] for automatic differentiation. As shown in the results, our FD decomposition significantly promotes efficiency in respect of both speed and memory usage, while the empirical approxima-

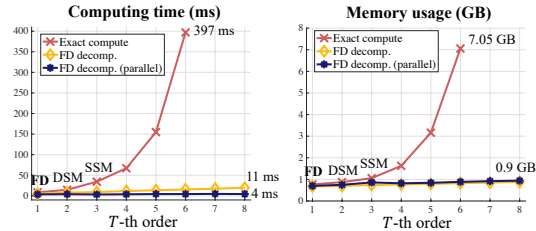

Figure 2: Computing time and memory usage for calculating the $T$-th order directional derivative.

tion error rates are kept within 1%. When we parallelize the FD decomposition, the computing time is almost a constant w.r.t. the order $T$, as long as there is enough GPU memory. In our experiments in Sec. 6, the computational efficiency is additionally validated on the FD-reformulated SM methods.

## 4 Application to score matching methods

Now we can instantiate $\mathcal{L}_\theta(x)$ in Eq. (8) as the log-density function $\log p_\theta(x)$ to reformulate the gradient-based SM methods. For unnormalized models $p_\theta(x) = \widetilde{p}_\theta(x)/Z_\theta$, the decomposition in Theorem 1 can naturally circumvent $Z_\theta$ by, e.g., $\log p_\theta(x + \alpha_k v) - \log p_\theta(x - \alpha_k v) = \log \widetilde{p}_\theta(x + \alpha_k v) - \log \widetilde{p}_\theta(x - \alpha_k v)$ where the partition function term cancels out, even without taking derivatives. Thus, the FD reformulations introduced in this section maintain the desirable property of their gradient-based counterparts of bypassing the intractable partition function. For simplicity, we set the random projection $v$ to be uniformly distributed as $p_\epsilon(v) = \mathcal{U}(\{v \in \mathbb{R}^d \,|\, \|v\| = \epsilon\})$, while our conclusions generally hold for other distributions of $v$ with bounded support sets.

**Finite-difference SSM.** For SSM, the scale factor is $C_v = \epsilon^2$ in Eq. (3). By instantiating $\mathcal{L}_\theta = \log p_\theta(x)$ in Eq. (8), we propose the finite-difference SSM (**FD-SSM**) objective as

$$
\begin{aligned}
\mathcal{J}_{\text{FD-SSM}}(\theta) = \frac{1}{\epsilon^2} \mathbb{E}_{p_{\text{data}}(x)} \mathbb{E}_{p_\epsilon(v)} \Big[ & \log p_\theta(x + v) + \log p_\theta(x - v) - 2 \log p_\theta(x) \\
& + \frac{1}{8} \left( \log p_\theta(x + v) - \log p_\theta(x - v) \right)^2 \Big] = \mathcal{J}_{\text{SSM}}(\theta) + o(\epsilon).
\end{aligned}
\tag{9}
$$

In Fig. 1, we intuitively illustrate the computational graph to better highlight the difference between the gradient-based objectives and their FD reformations, taking SSM as an example.

**Finite-difference DSM.** To construct the FD instantiation for DSM, we first cast the original objective in Eq. (2) into sliced Wasserstein distance [49] with random projection $v$ (detailed in Appendix B.4).

Then we can propose the finite-difference DSM (**FD-DSM**) objective as

$$\mathcal{J}_{\text{FD-DSM}}(\theta) = \frac{1}{4\epsilon^2} \mathbb{E}_{p_{\text{data}}(x)} \mathbb{E}_{p_\sigma(\widetilde{x}|x)} \mathbb{E}_{p_\epsilon(v)} \left[ \left( \log p_\theta(\widetilde{x}+v) - \log p_\theta(\widetilde{x}-v) + \frac{2v^\top(\widetilde{x}-x)}{\sigma^2} \right)^2 \right]. \quad (10)$$

It is easy to verify that $\mathcal{J}_{\text{FD-DSM}}(\theta) = \mathcal{J}_{\text{DSM}}(\theta) + o(\epsilon)$, and we can generalize FD-DSM to the cases with other noise distributions of $p_\sigma(\widetilde{x}|x)$ using similar instantiations of Eq. (8).

**Finite-difference SSMVR.** Our FD reformulation can also be used for *score-based generative models* [52, 57], where $s_\theta(x) : \mathbb{R}^d \to \mathbb{R}^d$ estimates $\nabla_x \log p_{\text{data}}(x)$ without modeling the likelihood by $p_\theta(x)$. In this case, we utilize the fact that $\mathbb{E}_{p_\epsilon(v)}\left[vv^\top\right] = \frac{\epsilon^2 I}{d}$ and focus on the objective of SSM with variance reduction (SSMVR) [58], where $\frac{1}{\epsilon^2}\mathbb{E}_{p_\epsilon(v)}[(v^\top s_\theta(x))^2] = \frac{1}{d}\|s_\theta(x)\|_2^2$ as

$$\mathcal{J}_{\text{SSMVR}}(\theta) = \mathbb{E}_{p_{\text{data}}(x)} \mathbb{E}_{p_\epsilon(v)} \left[ \frac{1}{\epsilon^2} v^\top \nabla_x s_\theta(x) v + \frac{1}{2d}\|s_\theta(x)\|_2^2 \right]. \quad (11)$$

If $s_\theta(x)$ is (element-wisely) twice-differentiable at $x$, we have the expansion that $s_\theta(x+v) + s_\theta(x-v) = 2s_\theta(x) + o(\epsilon)$ and $s_\theta(x+v) - s_\theta(x-v) = 2\nabla_x s_\theta(x)v + o(\epsilon^2)$. Then we can construct the finite-difference SSMVR (**FD-SSMVR**) for the score-based models as

$$\mathcal{J}_{\text{FD-SSMVR}}(\theta) = \mathbb{E}_{p_{\text{data}}(x)} \mathbb{E}_{p_\epsilon(v)} \left[ \frac{1}{8d}\|s_\theta(x+v) + s_\theta(x-v)\|_2^2 + \frac{1}{2\epsilon^2}\left(v^\top s_\theta(x+v) - v^\top s_\theta(x-v)\right) \right].$$

We can verify that $\mathcal{J}_{\text{FD-SSMVR}}(\theta) = \mathcal{J}_{\text{SSMVR}}(\theta) + o(\epsilon)$. Compared to the FD-SSM objective on the likelihood-based models, we only use two counterparts $s_\theta(x+v)$ and $s_\theta(x-v)$ in this instantiation.

## 5 Consistency under stochastic optimization

In practice, we usually apply mini-batch stochastic gradient descent (SGD) [51] to update the model parameters $\theta$. Thus beyond the expected $o(\epsilon)$ approximation error derived in Sec. 4, it is critical to formally verify the consistency between the FD-form objectives and their gradient-based counterparts under stochastic optimization. To this end, we establish a uniform convergence theorem for FD-SSM as an example, while similar proofs can be applied to other FD instantiations as detailed in Appendix B.5. A key insight is to show that the directions of $\nabla_\theta \mathcal{J}_{\text{FD-SSM}}(\theta)$ and $\nabla_\theta \mathcal{J}_{\text{SSM}}(\theta)$ are sufficiently aligned under SGD, as stated in Lemma 2:

**Lemma 2.** *(Uniform guarantee) Let $\mathcal{S}$ be the parameter space of $\theta$, $B$ be a bounded set in the space of $\mathbb{R}^d \times \mathcal{S}$, and $B_{\epsilon_0}$ be the $\epsilon_0$-neighbourhood of $B$ for certain $\epsilon_0 > 0$. Then under the condition that $\log p_\theta(x)$ is four times continuously differentiable w.r.t. $(x, \theta)$ and $\|\nabla_\theta \mathcal{J}_{\text{SSM}}(x, v; \theta)\|_2 > 0$ in the closure of $B_{\epsilon_0}$, we have $\forall \eta > 0, \exists \xi > 0$, such that*

$$\angle \left( \nabla_\theta \mathcal{J}_{\text{FD-SSM}}(x, v; \theta), \nabla_\theta \mathcal{J}_{\text{SSM}}(x, v; \theta) \right) < \eta \quad (12)$$

*uniformly holds for $\forall (x, \theta) \in B, v \in \mathbb{R}^d, \|v\|_2 = \epsilon < \min(\xi, \epsilon_0)$. Here $\angle(\cdot, \cdot)$ denotes the angle between two vectors. The arguments $x, v$ in the objectives indicate the losses at that point.*

Note that during the training process, we do not need to define a specific bounded set $B$ since our models are assumed to be globally differentiable in $\mathbb{R}^d \times \mathcal{S}$. This compact set only implicitly depends on the training process and the value of $\epsilon$. Based on Lemma 2 and other common assumptions in stochastic optimization [4], FD-SSM converges to a stationary point of SSM, as stated below:

**Theorem 2.** *(Consistency under SGD) Optimizing $\nabla_\theta \mathcal{J}_{\text{FD-SSM}}(\theta)$ with stochastic gradient descent, then the model parameters $\theta$ will converge to the stationary point of $\mathcal{J}_{\text{SSM}}(\theta)$ under the conditions including: (i) the assumptions for general stochastic optimization in Bottou et al. [4] hold; (ii) the differentiability assumptions in Lemma 2 hold; (iii) $\epsilon$ decays to zero during training.*

In the proof, we further show that the conditions (*i*) and (*ii*) largely overlap, and these assumptions are satisfied by the models described in the remark of Sec. 3.2. As to the condition (*iii*), we observe that in practice it is enough to set $\epsilon$ be a small constant during training, as shown in our experiments.

## 6 Experiments

In this section, we experiment on a diverse set of generative models, following the default settings in previous work [36, 57, 58].[2] It is worth clarifying that we use the same number of training iterations

Table 1: Results of the DKEF model on three UCI datasets. We report the negative log-likelihood (NLL) and the exact SM loss on the test set, as well as the training time per iteration. Under each algorithm, we train the DKEF model for 500 epochs with the batch size of 200.

| Algorithm | Parkinsons | | | RedWine | | | WhiteWine | | |
|---|---|---|---|---|---|---|---|---|---|
| | NLL | SM loss | Time | NLL | SM loss | Time | NLL | SM loss | Time |
| SSM | 14.52 | $-123.54$ | 110 ms | 13.34 | $-33.28$ | 113 ms | 14.13 | $-38.43$ | 105 ms |
| SSMVR | 13.26 | $-193.97$ | 111 ms | 13.13 | $-31.19$ | 106 ms | 13.63 | $-39.42$ | 111 ms |
| **FD-SSM** | 13.69 | $-138.72$ | **82.5 ms** | 13.06 | $-30.34$ | **82 ms** | 14.10 | $-32.84$ | **81.0 ms** |

Table 2: Results of deep EBMs on MNIST trained for 300K iterations with the batch size of 64. Here $\star$ indicates non-parallelized implementation of the FD objectives.

| Algorithm | SM loss | Time | Mem. |
|---|---|---|---|
| DSM | $-9.47 \times 10^4$ | 282 ms | 3.0 G |
| **FD-DSM$^\star$** | $-9.24 \times 10^4$ | 191 ms | 3.2 G |
| **FD-DSM** | $-9.27 \times 10^4$ | **162 ms** | 2.7 G |
| SSM | $-2.97 \times 10^7$ | 673 ms | 5.1 G |
| SSMVR | $-3.09 \times 10^7$ | 670 ms | 5.0 G |
| **FD-SSM$^\star$** | $-3.36 \times 10^7$ | 276 ms | 3.7 G |
| **FD-SSM** | $-3.33 \times 10^7$ | **230 ms** | 3.4 G |

Table 3: Results of the NICE model trained for 100 epochs with the batch size of 128 on MNIST. Here $^\dagger$ indicates $\sigma = 0.1$ [58] and $^{\dagger\dagger}$ indicates $\sigma = 1.74$ [53].

| Algorithm | SM loss | NLL | Time |
|---|---|---|---|
| Approx BP | $-2530 \pm 617$ | $1853 \pm 819$ | 55.3 ms |
| CP | $-2049 \pm 630$ | $1626 \pm 269$ | 73.6 ms |
| DSM$^\dagger$ | $-2820 \pm 825$ | $3398 \pm 1343$ | 35.8 ms |
| DSM$^{\dagger\dagger}$ | $-180 \pm 50$ | $3764 \pm 1583$ | 37.2 ms |
| SSM | $-2182 \pm 269$ | $2579 \pm 945$ | 59.6 ms |
| SSMVR | $-4943 \pm 3191$ | $6234 \pm 3782$ | 61.7 ms |
| **FD-SSM** | $-2425 \pm 100$ | $1647 \pm 306$ | **26.4 ms** |
| MLE | $-1236 \pm 525$ | $791 \pm 14$ | 24.3 ms |

for our FD methods as their gradient-based counterparts, while we report the time per iteration to exclude the compiling time. More implementation and definition details are in Appendix C.2.

## 6.1 Energy-based generative models

**Deep EBMs** utilize the capacity of neural networks to define unnormalized models. The backbone we use is an 18-layer ResNet [17] following Li et al. [36]. We validate our methods on six datasets including MNIST [33], Fashion-MNIST [69], CelebA [38], CIFAR-10 [28], SVHN [44], and ImageNet [7]. For CelebA and ImageNet, we adopt the officially cropped images and respectively resize to $32 \times 32$ and $128 \times 128$. The quantitative results on MNIST are given in Table 2. As shown, our FD formulations result in $2.9\times$ and $1.7\times$ speedup compared to the gradient-based SSM and DSM, respectively, with consistent SM losses. We simply set $\epsilon = 0.1$ to be a constant during training, since we find that the performance of our FD reformulations is insensitive to a wide value range of $\epsilon$. In Fig. 4 (a) and (b), we provide the loss curve of DSM / FD-DSM and SSM / FD-SSM w.r.t. time. As seen, FD-DSM can achieve the best model (lowest SM loss) faster, but eventually converges to higher loss compared to DSM. In contrast, when applying FD on SSM-based methods, the improvements are much more significant. This indicates that the random projection trick required by the FD formula is its main downside, which may outweigh the gain on efficiency for low-order computations.

As an additional evaluation of the learned model's performance, we consider two tasks using deep EBMs: the first one is **out-of-distribution detection**, where we follow previous work [6, 43] to use typicality as the detection metric (details in Appendix C.3), and report the AUC scores [18] and the training time per iteration in Table 4; the second one is **image generation**, where we apply annealed Langevin dynamics [36, 45, 67, 70] for inference and show the generated samples in the left of Fig. 3.

**Deep kernel exponential family (DKEF)** [68] is another unnormalized density estimator in the form of $\log \tilde{p}(x) = f(x) + \log p_0(x)$, with $p_0$ be the base measure, $f(x)$ defined as $\sum_{i=1}^{N} \sum_{j=1}^{N_j} k_i(x, z_j)$, where $N$ is the number of kernels, $k(\cdot, \cdot)$ is the Gaussian kernel function, and $z_j$ ($j = 0, \cdots, N_j$) are $N_j$ inducing points. The features are extracted using a neural network and the parameters of both the network and the kernel can be learned jointly using SM. Following the setting in Song et al. [58], we evaluate on three UCI datasets [2] and report the results in Table 1. As done for SSM, we calculate the tractable solution of the kernel method when training DKEF. The shared calculation leads to a relatively lower speed-up ratio of our FD method compared to the deep EBM case. For the choice of $\epsilon$, we found that the performances are insensitive to $\epsilon$: on the Parkinson dataset, the test NLLs and their corresponding $\epsilon$ are: $14.17(\epsilon = 0.1)$, $13.51(\epsilon = 0.05)$, $14.03(\epsilon = 0.02)$, $14.00(\epsilon = 0.01)$.

Table 4: Results of the out-of-distribution detection on deep EBMs. Training time per iteration and AUC scores ($M=2$ in typicality).

| Dataset | Algorithm | Time | SVHN | CIFAR | ImageNet |
|---------|-----------|------|------|-------|----------|
| SVHN | DSM | 673 ms | 0.49 | 1.00 | 0.99 |
| | **FD-DSM** | **305 ms** | 0.50 | 1.00 | 1.00 |
| CIFAR | DSM | 635 ms | 0.91 | 0.49 | 0.79 |
| | **FD-DSM** | **311 ms** | 0.92 | 0.51 | 0.81 |
| ImageNet | DSM | 1125 ms | 0.95 | 0.87 | 0.49 |
| | **FD-DSM** | **713 ms** | 0.95 | 0.89 | 0.49 |

Table 5: Results of the NCSN model trained for 200K iterations with 128 batch size on CIFAR-10. We report time per iteration and the FID scores.

| Algorithm | FID | Time | Mem. |
|-----------|-----|------|------|
| SSMVR | 41.2 | 865 ms | 6.4 G |
| **FD-SSMVR** | **39.5** | **575 ms** | 5.5 G |

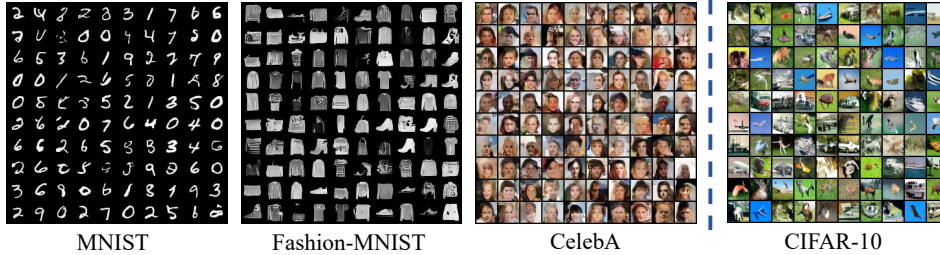

MNIST      Fashion-MNIST      CelebA      CIFAR-10

Figure 3: *Left.* The generated samples from deep EBMs trained by FD-DSM on MNIST, Fashion-MNIST and CelebA; *Right.* The generated samples from NCSN trained by FD-SSMVR on CIFAR-10.

## 6.2 Flow-based generative models

In addition to the unnormalized density estimators, SM methods can also be applied to flow-based models, whose log-likelihood functions are tractable and can be directly trained with MLE. Following Song et al. [58], we adopt the NICE [8] model and train it by minimizing the Fisher divergence using different approaches including approximate back-propagation (Approx BP) [25] and curvature propagation (CP) [39]. As in Table 3, FD-SSM achieves consistent results compared to SSM, while the training time is nearly comparable with the direct MLE, due to parallelization. The results are averaged over 5 runs except the SM based methods which are averaged over 10 runs. However, the variance is still large. We hypothesis that it is because the numerical stability of the baseline methods are relatively poor. In contrast, the variance of FD-SSM on the SM loss is much smaller, which shows better numerical stability of the shallower computational graphs induced by the FD decomposition.

## 6.3 Latent variable models with implicit encoders

SM methods can be also used in score estimation [35, 54, 61]. One particular application is on VAE [24] / WAE [64] with implicit encoders, where the gradient of the entropy term in the ELBO w.r.t. model parameters can be estimated (more details can be found in Song et al. [58] and Shi et al. [55]). We follow Song et al. [58] to evaluate VAE / WAE on both the MNIST and CelebA datasets using both SSMVR and FD-SSMVR. We report the results in Table 6. The reported training time only consists of the score estimation part, i.e., training the score model. As expected, the FD reformulation can improve computational efficiency without sacrificing the performance. The discussions concerned with other applications on the latent variable models can be found in Appendix B.6.

## 6.4 Score-based generative models

The noise conditional score network (NCSN) [57] trains a single score network $s_\theta(x, \sigma)$ to estimate the scores corresponding to all noise levels of $\sigma$. The noise level $\{\sigma_i\}_{i\in[10]}$ is a geometric sequence with $\sigma_1 = 1$ and $\sigma_{10} = 0.01$. When using the annealed Langevin dynamics for image generation, the number of iterations under each noise level is 100 with a uniform noise as the initial sample. As to the training approach of NCSN, Song and Ermon [57] mainly use DSM to pursue state-of-the-art performance, while we use SSMVR to demonstrate the efficiency of our FD reformulation. We train the models on the CIFAR-10 dataset with the batch size of 128 and compute the FID scores [19] on $50,000$ generated samples. We report the results in Table 5 and provide the generated samples in the right panel of Fig. 3. We also provide a curve in Fig. 4 (c) showing the FID scores (on 1,000 samples) during training. As seen, our FD methods can effectively learn different generative models.

Table 6: The results of training implicit encoders for VAE and WAE on the MNIST and CelebA datasets. The models are trained for 100K iterations with the batch size of 128.

| Model | Algorithm | MNIST | | CelebA | |
|---|---|---|---|---|---|
| | | NLL | Time | FID | Time |
| VAE | SSMVR | 89.58 | 5.04 ms | 62.76 | 14.9 ms |
| | **FD-SSMVR** | 88.96 | **3.98 ms** | 64.85 | **9.38 ms** |
| WAE | SSMVR | 90.45 | 0.55 ms | 54.28 | 1.30 ms |
| | **FD-SSMVR** | 90.66 | **0.39 ms** | 54.67 | **0.81 ms** |

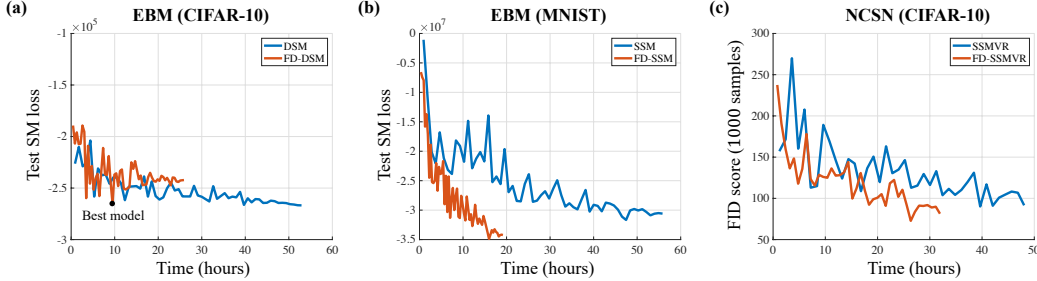

Figure 4: **(a)** Loss for DSM and FD-DSM; **(b)** Loss for SSM and FD-SSM; **(c)** FID scores on 1,000 samples (higher than those reported on 50,000 samples in Table 5) for SSMVR and FD-SSMVR.

# 7 Related work

In numerical analysis, the FD approaches play a central role in solving differential equations [60]. In machine learning, there have been related efforts devoted to leveraging the FD forms, either explicitly or implicitly. For a general scalar function $\mathcal{L}(x)$, we denote $H(x)$ as the Hessian matrix, $J(x)$ as the gradient, and $\sigma$ be a small value. LeCun [32] introduces a row-wise approximation of Hessian matrix as $H_k(x) \approx \frac{1}{\sigma}(J(x + \sigma e_k) - J(x))$, where $H_k$ represents the $k$-th row of Hessian matrix and $e_k$ is the $k$-th Euclidean basis vector. Rifai et al. [50] provide a FD approximation for the Frobenius norm of Hessian matrix as $\|H(x)\|_F^2 \approx \frac{1}{\sigma^2}\mathbb{E}[\|J(x + v) - J(x)\|_2^2]$, where $v \sim \mathcal{N}(0, \sigma^2 I)$ and the formulas is used to regularize the unsupervised auto-encoders. Møller [42] approximates the Hessian-vector product $H(x)v$ by calculating the directional FD as $H(x)v \approx \frac{1}{\sigma}(J(x + \sigma v) - J(x))$. Compared to our work, these previous methods mainly use the first-order terms $J(x)$ to approximate the second-order terms of $H(x)$, while we utilize the linear combinations of the original function $\mathcal{L}(x)$ to estimate high-order terms that exist in the Taylor's expansion, e.g., $v^\top H(x)v$.

As to the more implicit connections to FD, the minimum probability flow (MPF) [56] is a method for parameter estimation in probabilistic models. It is demonstrated that MPF can be connected to SM by a FD reformulation, where we provide a concise derivation in Appendix B.7. The noise-contrastive estimation (NCE) [14] train the unnormalized models by comparing the model distribution $p_\theta(x)$ with a noise distribution $p_n(x)$. It is proven that when we choose $p_n(x) = p_{\text{data}}(x + v)$ with a small vector $v$, i.e., $\|v\| = \epsilon$, the NCE objective can be equivalent to a FD approximation for the SSM objective as to an $o(1)$ approximation error rate after scaling [58]. In contrast, our FD-SSM method can achieve $o(\epsilon)$ approximation error with the same computational cost as NCE.

# 8 Conclusion

We propose to reformulate existing gradient-based SM methods using finite difference (FD), and theoretically and empirically demonstrate the consistency and computational efficiency of the FD-based training objectives. In addition to generative modeling, our generic FD decomposition can potentially be used in other applications involving higher-order derivatives. However, the price paid for this significant efficiency is that we need to work on the projected function in a certain direction, e.g., in DSM we need to first convert it into the slice Wasserstein distance and then apply the FD reformulation. This raises a trade-off between efficiency and variance in some cases.

## Broader Impact

This work proposes an efficient way to learn generative models and does not have a direct impact on society. However, by reducing the computation required for training unnormalized models, it may facilitate large-scale applications of, e.g., EBMs to real-world problems, which could have both positive (e.g., anomaly detection and denoising) and negative (e.g., deepfakes) consequences.

## Acknowledgements

This work was supported by the National Key Research and Development Program of China (No.2017YFA0700904), NSFC Projects (Nos. 61620106010, 62076145, U19B2034, U1811461), Beijing Academy of Artificial Intelligence (BAAI), Tsinghua-Huawei Joint Research Program, a grant from Tsinghua Institute for Guo Qiang, Tiangong Institute for Intelligent Computing, and the NVIDIA NVAIL Program with GPU/DGX Acceleration. C. Li was supported by the Chinese postdoctoral innovative talent support program and Shuimu Tsinghua Scholar.

## Footnotes

[1] Similar conclusions as in Lemma 1 and Theorem 1 were previously found in the Chapter 6.5 of Isaacson and Keller [22] under the univariate case, while we generalize them to the multivariate case.

[2]Our code is provided in https://github.com/taufikxu/FD-ScoreMatching.

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
