[Supplementary Material]

# A  Proofs

In this section we provide proofs for the conclusions in the main text.

## A.1  Proof of Lemma 1

If $\mathcal{L}_\theta(x)$ is $T$-times-differentiable at $x$, then according to the general form of multivariate Taylor's theorem [7], there is

$$\mathcal{L}_\theta(x + \gamma v) = \sum_{t=0}^{T} \gamma^t G_\theta^t(x, v, \epsilon) + o(\epsilon^T), \text{ where } G_\theta^t(x, v, \epsilon) = \left( \frac{\epsilon^t}{t!} \frac{\partial^t}{\partial v^t} \mathcal{L}_\theta(x) \right). \tag{1}$$

In order to extract the $T$-th component $G_\theta^t(x, v, \epsilon)$, we arbitrarily select a set of $T + 1$ different real values as $\{\gamma_i\}_{i \in [T+1]}$, and denote the induced Vandermonde matrix $V$ as

$$V = \begin{bmatrix} 1 & \gamma_1 & \gamma_1^2 & \cdots & \gamma_1^T \\ \vdots & \vdots & \vdots & \ddots & \vdots \\ 1 & \gamma_{(T+1)} & \gamma_{(T+1)}^2 & \cdots & \gamma_{(T+1)}^T \end{bmatrix}_{(T+1) \times (T+1)} , \text{ and } \boldsymbol{\beta} = \begin{pmatrix} \beta_1 \\ \vdots \\ \beta_{(T+1)} \end{pmatrix}_{(T+1) \times 1} , \tag{2}$$

where $\boldsymbol{\beta}$ is the vector of coefficients. The determinant of the Vandermonde matrix $V$ is $\det(V) = \prod_{i<j}(\gamma_j - \gamma_i) \neq 0$ since the values $\gamma_i$ are distinct. We consider the linear combination

$$\sum_{i=1}^{T+1} \beta_i \mathcal{L}_\theta(x + \gamma_i v) = \sum_{t=0}^{T} \phi_t G_\theta^t(x, v, \epsilon) + o(\epsilon^T), \text{ where } V^\top \boldsymbol{\beta} = \phi \in \mathbb{R}^{T+1}. \tag{3}$$

To eliminate the term $G_\theta^t(x, v, \epsilon)$ for any $t < T$ and keep the $T$-th order term $G_\theta^T(x, v, \epsilon)$, we just need to set the coefficient vector $\boldsymbol{\beta}$ be solution of $V^\top \boldsymbol{\beta} = \boldsymbol{e}_{(T+1)}$, where $\boldsymbol{e}_{(T+1)}$ is the one-hot vector of the $(T+1)$-th element. Then we have

$$\sum_{i=1}^{T+1} \beta_i \mathcal{L}_\theta(x + \gamma_i v) = G_\theta^T(x, v, \epsilon) + o(\epsilon^T) \Rightarrow \epsilon^T \frac{\partial^T}{\partial v^T} \mathcal{L}_\theta(x) = T! \sum_{i=1}^{T+1} \beta_i \mathcal{L}_\theta(x + \gamma_i v) + o(\epsilon^T). \tag{4}$$

$\square$

## A.2  Proof of Theorem 1

Let $K \in \mathbb{N}^+$, and $\{\alpha_k\}_{k \in [K]}$ be any set of $K$ different positive numbers, $\boldsymbol{\beta} = (\beta_1, \cdots, \beta_K) \in \mathbb{R}^K$ be a coefficient vector. Assuming that $\mathcal{L}_\theta(x)$ is $(T+1)$-times-differentiable at $x$. When $T = 2K$ is an even number, we select the coefficient set to be $\{\pm \alpha_1, \cdots, \pm \alpha_K\}$. Then we can construct the linear combination

$$\lambda \mathcal{L}_\theta(x) + \frac{1}{2} \sum_{k=1}^{K} \beta_k \alpha_k^{-2} \left[ \mathcal{L}_\theta(x + \alpha_k v) + \mathcal{L}_\theta(x - \alpha_k v) \right]$$

$$= \lambda \mathcal{L}_\theta(x) + \frac{1}{2} \sum_{k=1}^{K} \beta_k \alpha_k^{-2} \sum_{t=0}^{T+1} \left( 1 + (-1)^t \right) \alpha_k^t G_\theta^t(x, v, \epsilon) + o(\epsilon^{T+1}) \tag{5}$$

$$= \lambda \mathcal{L}_\theta(x) + \sum_{k=1}^{K} \beta_k \alpha_k^{-2} \sum_{t=0}^{K} \alpha_k^{2t} G_\theta^{2t}(x, v, \epsilon) + o(\epsilon^{T+1}),$$

where the second equation holds because $(1 + (-1)^t) = 0$ for any odd value of $t$. Note that there is $G_\theta^0(x, v, \epsilon) = \mathcal{L}_\theta(x)$, thus in order to eliminate the zero-order term, we let

$$\lambda = - \sum_{k=1}^{K} \beta_k \alpha_k^{-2}, \tag{6}$$

and then we can rewrite Eq. (5) as

$$
\frac{1}{2} \sum_{k=1}^{K} \beta_k \alpha_k^{-2} \left[ \mathcal{L}_\theta(x + \alpha_k v) + \mathcal{L}_\theta(x - \alpha_k v) - 2\mathcal{L}_\theta(x) \right]
$$

$$
= \sum_{k=1}^{K} \beta_k \alpha_k^{-2} \sum_{t=1}^{K} \alpha_k^{2t} G_\theta^{2t}(x, v, \epsilon) + o(\epsilon^{T+1}),
$$

$$
= \sum_{k=1}^{K} \beta_k \sum_{t=0}^{K-1} \alpha_k^{2t} G_\theta^{2t+2}(x, v, \epsilon) + o(\epsilon^{T+1}),
$$

(7)

Now in Eq. (7) we only need to eliminate the term $G_\theta^{2t+2}(x, v, \epsilon)$ for $t < K - 1$ and keep the term $G_\theta^{2K}(x, v, \epsilon)$, i.e., the $T$-th order term. we define the Vandermonde matrix $V$ generated by $\{\alpha_1^2, \cdots, \alpha_K^2\}$ as

$$
V = \begin{bmatrix} 1 & \alpha_1^2 & (\alpha_1^2)^2 & \cdots & (\alpha_1^2)^{K-1} \\ \vdots & \vdots & \vdots & \ddots & \vdots \\ 1 & \alpha_K^2 & (\alpha_K^2)^2 & \cdots & (\alpha_K^2)^{K-1} \end{bmatrix}_{K \times K}.
$$

(8)

It is easy to know that $V$ is non-singular as long as $\alpha_k$ are positive and different. Then if $\boldsymbol{\beta}$ is the solution of $V^\top \boldsymbol{\beta} = \boldsymbol{e}_K$ is the one-hot vector of the $K$-th element. Then we have

$$
\frac{1}{2} \sum_{k=1}^{K} \beta_k \alpha_k^{-2} \left[ \mathcal{L}_\theta(x + \alpha_k v) + \mathcal{L}_\theta(x - \alpha_k v) - 2\mathcal{L}_\theta(x) \right] = G_\theta^T(x, v, \epsilon) + o(\epsilon^{T+1}).
$$

(9)

Similarly when $T = 2K - 1$ is an odd number, we can construct the linear combination

$$
\frac{1}{2} \sum_{k=1}^{K} \beta_k \alpha_k^{-1} \left[ \mathcal{L}_\theta(x + \alpha_k v) - \mathcal{L}_\theta(x - \alpha_k v) \right]
$$

$$
= \frac{1}{2} \sum_{k=1}^{K} \beta_k \alpha_k^{-1} \sum_{t=0}^{T+1} \left( 1 - (-1)^t \right) \alpha_k^t G_\theta^t(x, v, \epsilon) + o(\epsilon^{T+1})
$$

$$
= \sum_{k=1}^{K} \beta_k \sum_{t=0}^{K-1} \alpha_k^{2t} G_\theta^{2t+1}(x, v, \epsilon) + o(\epsilon^{T+1}),
$$

(10)

where the second equation holds because $(1 - (-1)^t) = 0$ for any even value of $t$. Now we only need to eliminate the term $G_\theta^{2t+1}(x, v, \epsilon)$ for $t < K - 1$ and keep the term $G_\theta^{2K-1}(x, v, \epsilon)$, i.e., the $T$-th order term. Then if we still let $\boldsymbol{\beta}$ be the solution of $V_{\text{even}}^\top \boldsymbol{\beta} = \boldsymbol{e}_K$, we will have

$$
\frac{1}{2} \sum_{k=1}^{K} \beta_k \alpha_k^{-1} \left[ \mathcal{L}_\theta(x + \alpha_k v) - \mathcal{L}_\theta(x - \alpha_k v) \right] = G_\theta^T(x, v, \epsilon) + o(\epsilon^{T+1}).
$$

(11)

$\square$

### A.3 Proof of Lemma 2

We first investigate the gradient $\nabla_\theta \mathcal{J}_{\text{FD-SSM}}(x, v; \theta)$, whose elements consist of $\frac{\partial}{\partial \omega} \mathcal{J}_{\text{FD-SSM}}(x, v; \theta)$ for $\omega \in \theta$. Let $\overline{B}_{\epsilon_0}$ be the closure of $B_{\epsilon_0}$, then $\forall (x, \theta) \in B, t \in [-1, 1]$, there is $(x + t \cdot v, \theta) \in \overline{B}_{\epsilon_0}$ holds for any $v \in \mathbb{R}^d, \|v\|_2 = \epsilon < \epsilon_0$. It is easy to verify that $\overline{B}_{\epsilon_0}$ is a compact set. Since $\log p_\theta(x)$

is four times continuously differentiable in $\overline{B}_{\epsilon_0}$, we can obtain

$$
\begin{aligned}
&\frac{\partial}{\partial\omega}\log p_\theta(x+v) \\
=&\frac{\partial}{\partial\omega}\log p_\theta(x) + v^\top\nabla_x\frac{\partial}{\partial\omega}\log p_\theta(x) + \frac{1}{2}v^\top\nabla_x^2\frac{\partial}{\partial\omega}\log p_\theta(x)v + \sum_{|\boldsymbol{\alpha}|=3}R_{\boldsymbol{\alpha}}^\omega(x+v)\cdot v^{\boldsymbol{\alpha}} \\
=&\frac{\partial}{\partial\omega}\left[\log p_\theta(x) + v^\top\nabla_x\log p_\theta(x) + \frac{1}{2}v^\top\nabla_x^2\log p_\theta(x)v\right] + \sum_{|\boldsymbol{\alpha}|=3}R_{\boldsymbol{\alpha}}^\omega(x+v)\cdot v^{\boldsymbol{\alpha}},
\end{aligned}
\tag{12}
$$

where $|\boldsymbol{\alpha}| = \alpha_1 + \cdots + \alpha_d$, $\boldsymbol{\alpha}! = \alpha_1!\cdots\alpha_d!$, $v^{\boldsymbol{\alpha}} = v_1^{\alpha_1}\cdots v_d^{\alpha_d}$, and

$$
\begin{aligned}
R_{\boldsymbol{\alpha}}^\omega(x+v) &= \frac{|\boldsymbol{\alpha}|}{\boldsymbol{\alpha}!}\int_0^1(1-t)^{|\boldsymbol{\alpha}|-1}D^{\boldsymbol{\alpha}}\frac{\partial}{\partial\omega}\log p_\theta(x+t\cdot v)dt, \\
D^{\boldsymbol{\alpha}} &= \frac{\partial^{|\boldsymbol{\alpha}|}}{\partial x_1^{\alpha_1}\cdots\partial x_d^{\alpha_d}}.
\end{aligned}
\tag{13}
$$

Due to the continuity of the fourth-order derivatives of $\log p_\theta(x)$ on the compact set $\overline{B}_{\epsilon_0}$, we can obtain the uniform upper bound for $\forall(x,\theta)\in B, v\in\mathbb{R}^d, \|v\|_2 = \epsilon < \epsilon_0$ that

$$
|R_{\boldsymbol{\alpha}}^\omega(x+v)| \le U_\omega^+(\boldsymbol{\alpha}) \le \max_{\boldsymbol{\alpha}}U_\omega^+(\boldsymbol{\alpha}) = U_\omega^+.
\tag{14}
$$

So the remainder term in Eq. (12) has a upper bound as

$$
\left|\sum_{|\boldsymbol{\alpha}|=3}R_{\boldsymbol{\alpha}}^\omega(x+v)\cdot v^{\boldsymbol{\alpha}}\right| < d^3\epsilon^3 U_\omega^+,
\tag{15}
$$

where similar results also hold for $x-v$ and we represent the corresponding upper bound as $U_\omega^-$. Then we further have

$$
\begin{aligned}
&\frac{\partial}{\partial\omega}\left[\log p_\theta(x+v) + \log p_\theta(x-v) - 2\log p_\theta(x)\right] \\
=&\frac{\partial}{\partial\omega}v^\top\nabla_x^2\log p_\theta(x)v + \sum_{|\boldsymbol{\alpha}|=3}\left(R_{\boldsymbol{\alpha}}^\omega(x+v) + R_{\boldsymbol{\alpha}}^\omega(x-v)\right)\cdot v^{\boldsymbol{\alpha}}; \\
&\frac{\partial}{\partial\omega}\left[\log p_\theta(x+v) - \log p_\theta(x-v)\right] \\
=&2\frac{\partial}{\partial\omega}v^\top\nabla_x\log p_\theta(x) + \sum_{|\boldsymbol{\alpha}|=3}\left(R_{\boldsymbol{\alpha}}^\omega(x+v) - R_{\boldsymbol{\alpha}}^\omega(x-v)\right)\cdot v^{\boldsymbol{\alpha}}.
\end{aligned}
\tag{16}
$$

Similar for the expansion of $\log p_\theta(x+v)$, the remainder is

$$
R_{\boldsymbol{\alpha}}(x+v) = \frac{|\boldsymbol{\alpha}|}{\boldsymbol{\alpha}!}\int_0^1(1-t)^{|\boldsymbol{\alpha}|-1}D^{\boldsymbol{\alpha}}\log p_\theta(x+t\cdot v)dt
\tag{17}
$$

and we can obtain the uniform upper bound on the compact set $\overline{B}_{\epsilon_0}$ as

$$
|R_{\boldsymbol{\alpha}}(x+v)| \le U^+(\boldsymbol{\alpha}) \le \max_{\boldsymbol{\alpha}}U^+(\boldsymbol{\alpha}) = U^+.
\tag{18}
$$

We denote the bound for $R_{\boldsymbol{\alpha}}(x-v)$ as $U^-$ and further have

$$
\begin{aligned}
&\log p_\theta(x+v) - \log p_\theta(x-v) \\
=&2v^\top\nabla_x\log p_\theta(x) + \sum_{|\boldsymbol{\alpha}|=3}\left(R_{\boldsymbol{\alpha}}(x+v) - R_{\boldsymbol{\alpha}}(x-v)\right)\cdot v^{\boldsymbol{\alpha}}.
\end{aligned}
\tag{19}
$$

We denote $\Delta R_{\boldsymbol{\alpha}} = R_{\boldsymbol{\alpha}}(x+v) - R_{\boldsymbol{\alpha}}(x-v)$ and $\Delta R_{\boldsymbol{\alpha}}^{\omega,+} = R_{\boldsymbol{\alpha}}^\omega(x+v) + R_{\boldsymbol{\alpha}}^\omega(x-v)$ and $\Delta R_{\boldsymbol{\alpha}}^{\omega,-} = R_{\boldsymbol{\alpha}}^\omega(x+v) - R_{\boldsymbol{\alpha}}^\omega(x-v)$ for notation compactness. Thus for $\forall(x,\theta)\in B$ and $\|v\|_2 = \epsilon$,

we obtain the partial derivative of $\mathcal{J}_{\text{FD-SSM}}(x, v; \theta)$ as

$$
\begin{aligned}
&\frac{\partial}{\partial\omega}\mathcal{J}_{\text{FD-SSM}}(x, v; \theta) \\
=&\frac{1}{\epsilon^2}\left(v^\top\nabla_x\log p_\theta(x) + \frac{1}{2}\sum_{|\boldsymbol{\alpha}|=3}\Delta R_{\boldsymbol{\alpha}}\cdot v^{\boldsymbol{\alpha}}\right)\cdot\left(\frac{\partial}{\partial\omega}v^\top\nabla_x\log p_\theta(x) + \frac{1}{2}\sum_{|\boldsymbol{\alpha}|=3}\Delta R_{\boldsymbol{\alpha}}^{\omega,-}\cdot v^{\boldsymbol{\alpha}}\right) \\
&+ \frac{1}{\epsilon^2}\left(\frac{\partial}{\partial\omega}v^\top\nabla_x^2\log p_\theta(x)v + \sum_{|\boldsymbol{\alpha}|=3}\Delta R_{\boldsymbol{\alpha}}^{\omega,+}\cdot v^{\boldsymbol{\alpha}}\right) \\
=&\frac{1}{\epsilon^2}\left(v^\top\nabla_x\log p_\theta(x)\cdot\frac{\partial}{\partial\omega}v^\top\nabla_x\log p_\theta(x) + \frac{\partial}{\partial\omega}v^\top\nabla_x^2\log p_\theta(x)v\right) \\
&+ \frac{1}{\epsilon^2}\left(v^\top\nabla_x\log p_\theta(x)\cdot\frac{1}{2}\sum_{|\boldsymbol{\alpha}|=3}\Delta R_{\boldsymbol{\alpha}}^{\omega,-}\cdot v^{\boldsymbol{\alpha}} + \frac{\partial}{\partial\omega}v^\top\nabla_x\log p_\theta(x)\cdot\frac{1}{2}\sum_{|\boldsymbol{\alpha}|=3}\Delta R_{\boldsymbol{\alpha}}\cdot v^{\boldsymbol{\alpha}}\right) \\
&+ \frac{1}{\epsilon^2}\left(\left(\frac{1}{2}\sum_{|\boldsymbol{\alpha}|=3}\Delta R_{\boldsymbol{\alpha}}\cdot v^{\boldsymbol{\alpha}}\right)\cdot\left(\frac{1}{2}\sum_{|\boldsymbol{\alpha}|=3}\Delta R_{\boldsymbol{\alpha}}^{\omega,-}\cdot v^{\boldsymbol{\alpha}}\right) + \sum_{|\boldsymbol{\alpha}|=3}\Delta R_{\boldsymbol{\alpha}}^{\omega,+}\cdot v^{\boldsymbol{\alpha}}\right).
\end{aligned}
$$

Note that the first term in the above equals to $\frac{\partial}{\partial\omega}\mathcal{J}_{\text{SSM}}(x, v; \theta)$. Due to the continuity of the norm functions $\|\nabla_x\log p_\theta(x)\|_2$ and $\|\nabla_x\frac{\partial}{\partial\omega}\log p_\theta(x)\|_2$ on the compact set $\overline{B}_{\epsilon_0}$, we denote their upper bound as $G$ and $G_\omega$, respectively. Then we have $|v^\top\nabla_x\log p_\theta(x)| \leq \epsilon G$ and $\frac{\partial}{\partial\omega}v^\top\nabla_x\log p_\theta(x) = v^\top\nabla_x\frac{\partial}{\partial\omega}\log p_\theta(x) \leq \epsilon G_\omega$. Now we can derive the bound between the partial derivatives of FD-SSM and SSM as

$$
\begin{aligned}
&\left|\frac{\partial}{\partial\omega}\mathcal{J}_{\text{FD-SSM}}(x, v; \theta) - \frac{\partial}{\partial\omega}\mathcal{J}_{\text{SSM}}(x, v; \theta)\right| \\
<&\frac{1}{2\epsilon^2}\left(\epsilon G d^3\epsilon^3\Delta U_\omega + \epsilon G_\omega d^3\epsilon^3\Delta U + \frac{1}{2}\Delta U_\omega\Delta U d^6\epsilon^6 + 2\Delta U_\omega d^3\epsilon^3\right) \\
<&\epsilon\cdot\frac{1}{2}\left(Gd^2\Delta U_\omega + G_\omega d^2\Delta U + \frac{1}{2}\Delta U_\omega\Delta U d^3 + 2\Delta U_\omega d^3\right), \text{ holds when } \epsilon < \frac{1}{d},
\end{aligned}
\tag{20}
$$

where we denote $\Delta U = U^+ + U^-$ and $\Delta U_\omega = U_\omega^+ + U_\omega^-$. By setting $\epsilon_0 < \frac{1}{d}$, we can omit the condition $\epsilon < \frac{1}{d}$ since $\epsilon < \epsilon_0 = \min(\epsilon_0, \frac{1}{d})$. Note that the condition $\epsilon < \frac{1}{d}$ can be generalize to, e.g., $\epsilon < 1$ without changing our conclusions. Then it is easy to show that

$$
\begin{aligned}
&\|\nabla_\theta\mathcal{J}_{\text{FD-SSM}}(x, v; \theta) - \nabla_\theta\mathcal{J}_{\text{SSM}}(x, v; \theta)\|_2 \\
\leq&\dim(\mathcal{S})\cdot\max_{\omega\in\theta}\left|\frac{\partial}{\partial\omega}\mathcal{J}_{\text{FD-SSM}}(x, v; \theta) - \frac{\partial}{\partial\omega}\mathcal{J}_{\text{SSM}}(x, v; \theta)\right| \\
<&\epsilon\cdot\dim(\mathcal{S})\cdot\max_{\omega\in\theta}M_\omega, \\
&\text{where } M_\omega = \frac{1}{2}\left(Gd^2\Delta U_\omega + G_\omega d^2\Delta U + \frac{1}{2}\Delta U_\omega\Delta U d^3 + 2\Delta U_\omega d^3\right).
\end{aligned}
\tag{21}
$$

Just to emphasize here, the bound above uniformly holds for $\forall(x, \theta) \in B$ and $v \in \mathbb{R}^d, \|v\|_2 = \epsilon < \epsilon_0$. We have the simple fact that give two vectors $a$ and $b$, if there is $\|a - b\|_2 < \|b\|_2$, then their angle is $\angle(a, b) \leq \arcsin(\|a - b\|_2/\|b\|_2)$. So finally we can derive the angle

$$
\begin{aligned}
&\angle\left(\nabla_\theta\mathcal{J}_{\text{FD-SSM}}(x, v; \theta), \nabla_\theta\mathcal{J}_{\text{SSM}}(x, v; \theta)\right) \\
\leq&\arcsin\left(\frac{\|\nabla_\theta\mathcal{J}_{\text{FD-SSM}}(x, v; \theta) - \nabla_\theta\mathcal{J}_{\text{SSM}}(x, v; \theta)\|_2}{\|\nabla_\theta\mathcal{J}_{\text{SSM}}(x, v; \theta)\|_2}\right) \\
\leq&\arcsin\left(\frac{\epsilon\cdot\dim(\mathcal{S})\cdot\max_{\omega\in\theta}M_\omega}{\min_{(x,\theta)\in B, \|v\|_2<\epsilon_0}\|\nabla_\theta\mathcal{J}_{\text{SSM}}(x, v; \theta)\|_2}\right) \\
<&\eta,
\end{aligned}
\tag{22}
$$

where $\min_{(x,\theta)\in B, \|v\|_2 < \epsilon_0} \|\nabla_\theta \mathcal{J}_{\text{SSM}}(x, v; \theta)\|_2$ must exist and larger than 0 due to the continuity of $\nabla_\theta \mathcal{J}_{\text{SSM}}(x, v; \theta)$ and the condition that $\|\nabla_\theta \mathcal{J}_{\text{SSM}}(x, v; \theta)\|_2 > 0$ on the compact set. So we only need to choose $\xi$ as

$$\xi = \frac{\sin \eta \cdot \min_{(x,\theta)\in B, \|v\|_2 < \epsilon_0} \|\nabla_\theta \mathcal{J}_{\text{SSM}}(x, v; \theta)\|_2}{\dim(\mathcal{S}) \cdot \max_{\omega \in \theta} M_\omega}. \tag{23}$$

When we choose $\|v\|_2 = \epsilon < \min(\epsilon_0, \xi)$, we can guarantee the angle between $\nabla_\theta \mathcal{J}_{\text{FD-SSM}}(x, v; \theta)$ and $\nabla_\theta \mathcal{J}_{\text{SSM}}(x, v; \theta)$ to be uniformly less than $\eta$ on $B$. □

### A.4   Proof of Theorem 2

We consider in the compact set $\overline{B}_{\epsilon_0}$ defined in Lemma 2. The assumptions for general stochastic optimization include:

- **(i)** The condition of Corollary 4.12 in Bottou et al. [1]: $\mathcal{J}_{\text{FD-SSM}}(\theta)$ is twice-differentiable with $\theta$;
- **(ii)** The Assumption 4.1 in Bottou et al. [1]: the gradient $\nabla_\theta \mathcal{J}_{\text{FD-SSM}}(\theta)$ is Lipschitz;
- **(iii)** The Assumption 4.3 in Bottou et al. [1]: the first and second moments of the stochastic gradients are bounded by the expected gradients;
- **(iv)** The stochastic step size $\alpha_k$ satisfies the diminishing condition in Bottou et al. [1]: $\sum_{k=1}^{\infty} \alpha_k = \infty, \sum_{k=1}^{\infty} \alpha_k^2 < \infty$;
- **(v)** The condition of Lemma 2 holds in each step $k$ of stochastic gradient update.

Note that the condition **(v)** only holds in the compact set $\overline{B}_{\epsilon_0}$, but we can choose it to be large enough to contain $(x, \theta_k), x \sim p(x)$, as well as containing the neighborhood of stationary points of $\nabla_\theta \mathcal{J}_{\text{SSM}}(\theta)$. These can be achieved by setting $\epsilon \to 0$. Thus we have

$$\lim_{k \to \infty, \epsilon \to 0} \mathbb{E}\left[\|\nabla_\theta \mathcal{J}_{\text{SSM}}(\theta_k)\|_2\right] = 0. \tag{24}$$

This means that stochastically optimizing the FD-SSM objective can make the parameters $\theta$ converge to the stationary point of the SSM objective when $\epsilon \to 0$. □

## B   Extended conclusions

In this section we provide extended and supplementary conclusions for the main text.

### B.1   Parallel computing on dependent operations

For the dependent operations like those in the gradient-based SM methods, it is possible to execute them on different devices via asynchronous parallelism [9]. However, this asynchronous parallelization needs to perform across different data batches, requires complex design on the synchronization mechanism, and could introduce extra bias when updating the model parameters. These difficulties usually outweigh the gain from paralleling the operations in the gradient-based SM methods. In contrast, for our FD-based SM methods, the decomposed independent operations can be easily executed in a synchronous manner, which is further compatible with data or model parallelism.

### B.2   Scaling the projection vector in training objectives

Below we explain why the scale of the random projection $v$ will not affect the training of SM objectives. For the original SM objective, we have

$$\begin{aligned} \mathcal{J}_{\text{SM}}(\theta) &= \mathbb{E}_{p_{\text{data}}(x)} \left[ \text{tr}(\nabla_x^2 \log p_\theta(x)) + \frac{1}{2}\|\nabla_x \log p_\theta(x)\|_2^2 \right] \\ &= \mathbb{E}_{p_{\text{data}}(x)} \left[ \sum_{i=1}^{d} \boldsymbol{e}_i^\top \nabla_x^2 \log p_\theta(x) \boldsymbol{e}_i + \frac{1}{2} \sum_{i=1}^{d} \left(\boldsymbol{e}_i^\top \nabla_x \log p_\theta(x)\right)^2 \right]. \end{aligned} \tag{25}$$

When we scale the basis vector $e_i$ with a small value $\epsilon'$, i.e., $e_i \to \epsilon' e_i$, we have

$$
\mathbb{E}_{p_{\text{data}}(x)} \left[ \sum_{i=1}^{d} (\epsilon' e_i)^\top \nabla_x^2 \log p_\theta(x)(\epsilon' e_i) + \frac{1}{2} \sum_{i=1}^{d} \left( (\epsilon' e_i)^\top \nabla_x \log p_\theta(x) \right)^2 \right]
$$
$$
= \epsilon'^2 \mathbb{E}_{p_{\text{data}}(x)} \left[ \sum_{i=1}^{d} e_i^\top \nabla_x^2 \log p_\theta(x) e_i + \frac{1}{2} \sum_{i=1}^{d} \left( e_i^\top \nabla_x \log p_\theta(x) \right)^2 \right] = \epsilon'^2 \mathcal{J}_{\text{SM}}(\theta). \tag{26}
$$

Thus we can simply divide the objective by $\epsilon'^2$ to recover the original SM objective $\mathcal{J}_{\text{SM}}(\theta)$. Similarly, for the DSM objective we have

$$
\mathcal{J}_{\text{DSM}}(\theta) = \frac{1}{d} \mathbb{E}_{p_{\text{data}}(x)} \mathbb{E}_{p_\sigma(\widetilde{x}|x)} \left[ \left\| \nabla_{\widetilde{x}} \log p_\theta(\widetilde{x}) + \frac{\widetilde{x} - x}{\sigma^2} \right\|_2^2 \right]
$$
$$
= \frac{1}{d} \mathbb{E}_{p_{\text{data}}(x)} \mathbb{E}_{p_\sigma(\widetilde{x}|x)} \left[ \sum_{i=1}^{d} \left( e_i^\top \nabla_{\widetilde{x}} \log p_\theta(\widetilde{x}) + \frac{e_i^\top (\widetilde{x} - x)}{\sigma^2} \right)^2 \right]. \tag{27}
$$

When we scale the basis vector $e_i$ with a small value $\epsilon'$, i.e., $e_i \to \epsilon' e_i$, we also have

$$
\frac{1}{d} \mathbb{E}_{p_{\text{data}}(x)} \mathbb{E}_{p_\sigma(\widetilde{x}|x)} \left[ \sum_{i=1}^{d} \left( (\epsilon' e_i)^\top \nabla_{\widetilde{x}} \log p_\theta(\widetilde{x}) + \frac{(\epsilon' e_i)^\top (\widetilde{x} - x)}{\sigma^2} \right)^2 \right]
$$
$$
= \frac{\epsilon'^2}{d} \mathbb{E}_{p_{\text{data}}(x)} \mathbb{E}_{p_\sigma(\widetilde{x}|x)} \left[ \sum_{i=1}^{d} \left( e_i^\top \nabla_{\widetilde{x}} \log p_\theta(\widetilde{x}) + \frac{e_i^\top (\widetilde{x} - x)}{\sigma^2} \right)^2 \right] = \epsilon'^2 \mathcal{J}_{\text{DSM}}(\theta). \tag{28}
$$

Thus we can divide by $\epsilon'^2$ to recover the DSM objective $\mathcal{J}_{\text{DSM}}(\theta)$. Finally as to SSM, we have

$$
\mathcal{J}_{\text{SSM}}(\theta) = \frac{1}{C_v} \mathbb{E}_{p_{\text{data}}(x)} \mathbb{E}_{p_v(v)} \left[ v^\top \nabla_x^2 \log p_\theta(x) v + \frac{1}{2} \left( v^\top \nabla_x \log p_\theta(x) \right)^2 \right]. \tag{29}
$$

When we scale the random projection $v$ with a small value $\epsilon'$, i.e., $v \to \epsilon' v$, we should not that the adaptive factor $C_v$ will also be scaled to $\epsilon'^2 C_v$, then we can derive

$$
\frac{1}{\epsilon'^2 C_v} \mathbb{E}_{p_{\text{data}}(x)} \mathbb{E}_{p_v(v)} \left[ (\epsilon' v)^\top \nabla_x^2 \log p_\theta(x)(\epsilon' v) + \frac{1}{2} \left( (\epsilon' v)^\top \nabla_x \log p_\theta(x) \right)^2 \right] = \mathcal{J}_{\text{SSM}}(\theta). \tag{30}
$$

This indicates that the SSM objective is already invariant to the scaling of $v$. It is trivial to also divide similar factors as $C_v$ in SM and DSM to result in similarly invariant objectives.

## B.3 Mild regularity conditions for the FD-based SM methods

The mild conditions for the original gradient-based SM methods [5, 16] include: (i) $p_{\text{data}}(x)$ and $p_\theta(x)$ are both twice-differentiable on $\mathbb{R}^d$; (ii) $\mathbb{E}_{p_{\text{data}}(x)}[\|\nabla_x \log p_\theta(x)\|_2^2]$ and $\mathbb{E}_{p_{\text{data}}(x)}[\|\nabla_x \log p_{\text{data}}(x)\|_2^2]$ are finite for any $\theta$; (iii) There is $\lim_{\|x\| \to \infty} p_{\text{data}}(x) \nabla_x \log p_\theta(x) = 0$ holds for any $\theta$. Here we provide two additional regularity conditions which are sufficient to guarantee the $o(\epsilon)$ or $\mathcal{O}(\epsilon^2)$ approximation error of FD-SSM and FD-DSM: (iv) $p_\theta(x)$ is four-times continuously differentiable on $\mathbb{R}^d$; (v) There is $\mathbb{E}_{p_{\text{data}}(x)}[|D^{\boldsymbol{\alpha}} \log p_\theta(x)|] < \infty$ holds for any $\theta$ and $|\boldsymbol{\alpha}| = 4$, where $D$ and $\boldsymbol{\alpha}$ are defined in Eq. (13). The proof is almost the same as it for Theorem 1 under Lagrange's remainder.

**Remark.** Note that the condition (iv) holds when we apply, e.g., average pooling layers and Softplus activation in the neural network models, while the condition (v) always holds as long as the support set of $p_{\text{data}}(x)$ is bounded, e.g., for RGB-based image tasks there is $x \in [0, 255]^d$.

## B.4 DSM under sliced Wasserstein distance

To construct the FD instantiation for DSM, we first cast the original objective of DSM into sliced Wasserstein distance [13] with random projection $v$. Since there is $\mathbb{E}_{p_\epsilon(v)} \left[ vv^\top \right] = \frac{\epsilon^2 I}{d}$, we can rewrite the objective of DSM with Gaussian noise distribution as

$$
\mathcal{J}_{\text{DSM}}(\theta) = \frac{1}{\epsilon^2} \mathbb{E}_{p_{\text{data}}(x)} \mathbb{E}_{p_\sigma(\widetilde{x}|x)} \mathbb{E}_{p_\epsilon(v)} \left[ \left( v^\top \nabla_{\widetilde{x}} \log p_\theta(\widetilde{x}) + \frac{v^\top (\widetilde{x} - x)}{\sigma^2} \right)^2 \right]. \tag{31}
$$

In this case, there is $\frac{v^\top(\widetilde{x}-x)}{\sigma^2} = \mathcal{O}(\epsilon)$ with high probability, thus we can approximate $v^\top \nabla_{\widetilde{x}} \log p_\theta(\widetilde{x})$ according to our FD decomposition.

## B.5 Consistency between DSM and FD-DSM

**Theorem\* 1.** *Let $\mathcal{S}$ be the parameter space of $\theta$, $B$ be a bounded set in the space of $\mathbb{R}^d \times \mathcal{S}$, and $B_{\epsilon_0}$ is the $\epsilon_0$-neighbourhood of $B$ for certain $\epsilon_0 > 0$. Then under the condition that $\log p_\theta(\widetilde{x})$ is three times continuously differentiable w.r.t. $(\widetilde{x}, \theta)$ and $\|\nabla_\theta \mathcal{J}_{\mathrm{DSM}}(x, \widetilde{x}, v; \theta)\|_2 > 0$ in the closure of $B_{\epsilon_0}$, we have $\forall \eta > 0, \exists \xi > 0$, such that*

$$\angle\left(\nabla_\theta \mathcal{J}_{\mathrm{FD\text{-}DSM}}(x, \widetilde{x}, v; \theta), \nabla_\theta \mathcal{J}_{\mathrm{DSM}}(x, \widetilde{x}, v; \theta)\right) < \eta \tag{32}$$

*uniformly holds for $\forall (\widetilde{x}, \theta) \in B, v \in \mathbb{R}^d, \|v\|_2 = \epsilon < \min(\xi, \epsilon_0)$ and $x$ in any bounded subset of $\mathbb{R}^d$. Here $\angle(\cdot, \cdot)$ denotes the angle between two vectors. The arguments $x, \widetilde{x}, v$ in the objectives indicate the losses at that point.*

*Proof.* Following the routines and notations in the proof of Lemma 2, we investigate the gradient $\nabla_\theta \mathcal{J}_{\mathrm{FD\text{-}DSM}}(x, \widetilde{x}, v; \theta)$, whose elements consist of $\frac{\partial}{\partial \omega} \mathcal{J}_{\mathrm{FD\text{-}DSM}}(x, \widetilde{x}, v; \theta)$ for $\omega \in \theta$. When $\log p_\theta(\widetilde{x})$ is three-times-differentiable in $\overline{B}_{\epsilon_0}$, we can obtain

$$
\begin{aligned}
&\frac{\partial}{\partial \omega} \log p_\theta(\widetilde{x} + v) \\
&= \frac{\partial}{\partial \omega} \log p_\theta(\widetilde{x}) + v^\top \nabla_{\widetilde{x}} \frac{\partial}{\partial \omega} \log p_\theta(\widetilde{x}) + \frac{1}{2} v^\top \nabla_{\widetilde{x}}^2 \frac{\partial}{\partial \omega} \log p_\theta(\widetilde{x}) v + \sum_{|\boldsymbol{\alpha}|=3} R_{\boldsymbol{\alpha}}^\omega(\widetilde{x} + v) \cdot v^{\boldsymbol{\alpha}} \\
&= \frac{\partial}{\partial \omega} \left[ \log p_\theta(\widetilde{x}) + v^\top \nabla_{\widetilde{x}} \log p_\theta(\widetilde{x}) + \frac{1}{2} v^\top \nabla_{\widetilde{x}}^2 \log p_\theta(\widetilde{x}) v \right] + \sum_{|\boldsymbol{\alpha}|=3} R_{\boldsymbol{\alpha}}^\omega(\widetilde{x} + v) \cdot v^{\boldsymbol{\alpha}},
\end{aligned} \tag{33}
$$

$$\text{where } R_{\boldsymbol{\alpha}}^\omega(\widetilde{x} + v) = \frac{|\boldsymbol{\alpha}|}{\boldsymbol{\alpha}!} \int_0^1 (1-t)^{|\boldsymbol{\alpha}|-1} D^{\boldsymbol{\alpha}} \frac{\partial}{\partial \omega} \log p_\theta(\widetilde{x} + t \cdot v) dt,$$

Then we can further obtain that

$$\frac{\partial}{\partial \omega}\left[\log p_\theta(\widetilde{x}+v) - \log p_\theta(\widetilde{x}-v)\right] = 2\frac{\partial}{\partial \omega} v^\top \nabla_{\widetilde{x}} \log p_\theta(\widetilde{x}) + \sum_{|\boldsymbol{\alpha}|=3} \left(R_{\boldsymbol{\alpha}}^\omega(\widetilde{x}+v) - R_{\boldsymbol{\alpha}}^\omega(\widetilde{x}-v)\right) \cdot v^{\boldsymbol{\alpha}}.$$

Due to the continuity of $R_{\boldsymbol{\alpha}}^\omega(\widetilde{x} + v)$ and $R_{\boldsymbol{\alpha}}^\omega(\widetilde{x} - v)$ on the compact set $\overline{B}_{\epsilon_0}$, they have the uniform absolute upper bounds $U_\omega^+$ and $U_\omega^-$, respectively. Similarly, we have

$$\log p_\theta(\widetilde{x}+v) - \log p_\theta(\widetilde{x}-v) = 2v^\top \nabla_{\widetilde{x}} \log p_\theta(\widetilde{x}) + \sum_{|\boldsymbol{\alpha}|=3} \left(R_{\boldsymbol{\alpha}}(\widetilde{x}+v) - R_{\boldsymbol{\alpha}}(\widetilde{x}-v)\right) \cdot v^{\boldsymbol{\alpha}},$$

where the uniform absolute upper bounds for $R_{\boldsymbol{\alpha}}(\widetilde{x}+v)$ and $R_{\boldsymbol{\alpha}}(\widetilde{x}-v)$ are $U^+$ and $U^-$, respectively. Besides, note that the terms $\frac{x-\widetilde{x}}{\sigma}$ in the DSM / FD-DSM objectives are independent of $\theta$. We denote $\Delta R_{\boldsymbol{\alpha}} = R_{\boldsymbol{\alpha}}(\widetilde{x}+v) - R_{\boldsymbol{\alpha}}(\widetilde{x}-v)$ and $\Delta R_{\boldsymbol{\alpha}}^\omega = R_{\boldsymbol{\alpha}}^\omega(\widetilde{x}+v) - R_{\boldsymbol{\alpha}}^\omega(\widetilde{x}-v)$ for notation compactness. Thus for $\forall (\widetilde{x}, \theta) \in B$ and $\|v\|_2 = \epsilon, x \in \mathbb{R}^d$, we can obtain the partial derivative of $\mathcal{J}_{\mathrm{FD\text{-}DSM}}(x, \widetilde{x}, v; \theta)$ as

$$
\begin{aligned}
&\frac{\partial}{\partial \omega} \mathcal{J}_{\mathrm{FD\text{-}DSM}}(x, \widetilde{x}, v; \theta) \\
&= \frac{1}{2\epsilon^2}\left(2v^\top \nabla_{\widetilde{x}} \log p_\theta(\widetilde{x}) + \sum_{|\boldsymbol{\alpha}|=3} \Delta R_{\boldsymbol{\alpha}} \cdot v^{\boldsymbol{\alpha}} + \frac{2v^\top(\widetilde{x}-x)}{\sigma^2}\right) \cdot \left(2\frac{\partial}{\partial \omega} v^\top \nabla_{\widetilde{x}} \log p_\theta(\widetilde{x}) + \sum_{|\boldsymbol{\alpha}|=3} \Delta R_{\boldsymbol{\alpha}}^\omega \cdot v^{\boldsymbol{\alpha}}\right) \\
&= \frac{1}{\epsilon^2} \frac{\partial}{\partial \omega}\left(v^\top \nabla_{\widetilde{x}} \log p_\theta(\widetilde{x}) + \frac{v^\top(\widetilde{x}-x)}{\sigma^2}\right)^2 + \frac{1}{2\epsilon^2}\left(\sum_{|\boldsymbol{\alpha}|=3} \Delta R_{\boldsymbol{\alpha}}^\omega \cdot v^{\boldsymbol{\alpha}}\right) \cdot \left(\sum_{|\boldsymbol{\alpha}|=3} \Delta R_{\boldsymbol{\alpha}} \cdot v^{\boldsymbol{\alpha}}\right) \\
&\quad + \frac{1}{\epsilon^2}\left(\left(v^\top \nabla_{\widetilde{x}} \log p_\theta(\widetilde{x}) + \frac{v^\top(\widetilde{x}-x)}{\sigma^2}\right) \sum_{|\boldsymbol{\alpha}|=3} \Delta R_{\boldsymbol{\alpha}}^\omega \cdot v^{\boldsymbol{\alpha}} + \frac{\partial}{\partial \omega} v^\top \nabla_{\widetilde{x}} \log p_\theta(\widetilde{x}) \sum_{|\boldsymbol{\alpha}|=3} \Delta R_{\boldsymbol{\alpha}} \cdot v^{\boldsymbol{\alpha}}\right),
\end{aligned}
$$

where the first term equals to $\frac{\partial}{\partial\omega}\mathcal{J}_{\text{DSM}}(x,\widetilde{x},v;\theta)$. Due to the continuity of the norm functions $\|\nabla_{\widetilde{x}}\log p_\theta(\widetilde{x})\|_2$ and $\|\nabla_{\widetilde{x}}\frac{\partial}{\partial\omega}\log p_\theta(\widetilde{x})\|_2$ on the compact set $\overline{B}_{\epsilon_0}$, we denote their upper bound as $G$ and $G_\omega$, respectively. Then we have $|v^\top\nabla_{\widetilde{x}}\log p_\theta(\widetilde{x})| \leq \epsilon G$ and $\frac{\partial}{\partial\omega}v^\top\nabla_{\widetilde{x}}\log p_\theta(\widetilde{x}) = v^\top\nabla_{\widetilde{x}}\frac{\partial}{\partial\omega}\log p_\theta(\widetilde{x}) \leq \epsilon G_\omega$. Besides, since $\widetilde{x}$ and $x$ both come from bounded sets, we have an upper bound of $v^\top(\widetilde{x}-x) \leq \epsilon\sigma^2 G_x$. Now we can derive the bound between the partial derivatives of FD-DSM and DSM as

$$
|\frac{\partial}{\partial\omega}\mathcal{J}_{\text{FD-DSM}}(x,\widetilde{x},v;\theta) - \frac{\partial}{\partial\omega}\mathcal{J}_{\text{DSM}}(x,\widetilde{x},v;\theta)|
$$
$$
<\frac{1}{\epsilon^2}\left(\frac{1}{2}\Delta U\Delta U_\omega d^6\epsilon^6 + (G+G_x)\Delta U_\omega d^3\epsilon^3 + G_\omega\Delta U d^3\epsilon^3\right) \tag{34}
$$
$$
<\epsilon\cdot\left(\frac{1}{2}\Delta U\Delta U_\omega d^3 + (G+G_x)\Delta U_\omega d^3 + G_\omega\Delta U d^3\right), \text{ holds when } \epsilon < \frac{1}{d},
$$

where we denote $\Delta U = U^+ + U^-$ and $\Delta U_\omega = U_\omega^+ + U_\omega^-$. By setting $\epsilon_0 < \frac{1}{d}$, we can omit the condition $\epsilon < \frac{1}{d}$ since $\epsilon < \epsilon_0 = \min(\epsilon_0,\frac{1}{d})$. Note that the condition $\epsilon < \frac{1}{d}$ can be generalize to, e.g., $\epsilon < 1$ without changing our conclusions. Then it is easy to show that

$$
\|\nabla_\theta\mathcal{J}_{\text{FD-DSM}}(x,\widetilde{x},v;\theta) - \nabla_\theta\mathcal{J}_{\text{DSM}}(x,\widetilde{x},v;\theta)\|_2
$$
$$
\leq\dim(\mathcal{S})\cdot\max_{\omega\in\theta}\left|\frac{\partial}{\partial\omega}\mathcal{J}_{\text{FD-DSM}}(x,\widetilde{x},v;\theta) - \frac{\partial}{\partial\omega}\mathcal{J}_{\text{DSM}}(x,\widetilde{x},v;\theta)\right| \tag{35}
$$
$$
<\epsilon\cdot\dim(\mathcal{S})\cdot\max_{\omega\in\theta}M_\omega,
$$
$$
\text{where } M_\omega = \frac{1}{2}\Delta U\Delta U_\omega d^3 + (G+G_x)\Delta U_\omega d^3 + G_\omega\Delta U d^3.
$$

Just to emphasize here, the bound above uniformly holds for $\forall(\widetilde{x},\theta)\in B$ and $v\in\mathbb{R}^d, \|v\|_2 = \epsilon < \epsilon_0$ and $x$ from any bounded set ($x$ is inherently bounded when we consider, e.g., pixel input space). So finally we can derive the angle

$$
\angle\left(\nabla_\theta\mathcal{J}_{\text{FD-DSM}}(x,\widetilde{x},v;\theta),\nabla_\theta\mathcal{J}_{\text{DSM}}(x,\widetilde{x},v;\theta)\right)
$$
$$
\leq\arcsin\left(\frac{\|\nabla_\theta\mathcal{J}_{\text{FD-DSM}}(x,\widetilde{x},v;\theta) - \nabla_\theta\mathcal{J}_{\text{DSM}}(x,\widetilde{x},v;\theta)\|_2}{\|\nabla_\theta\mathcal{J}_{\text{DSM}}(x,\widetilde{x},v;\theta)\|_2}\right)
$$
$$
\leq\arcsin\left(\frac{\epsilon\cdot\dim(\mathcal{S})\cdot\max_{\omega\in\theta}M_\omega}{\min_{(\widetilde{x},\theta)\in B,\|v\|_2<\epsilon_0}\|\nabla_\theta\mathcal{J}_{\text{DSM}}(x,\widetilde{x},v;\theta)\|_2}\right) \tag{36}
$$
$$
<\eta,
$$

where $\min_{(\widetilde{x},\theta)\in B,\|v\|_2<\epsilon_0}\|\nabla_\theta\mathcal{J}_{\text{DSM}}(x,\widetilde{x},v;\theta)\|_2$ must exist and larger than 0 due to the continuity of $\nabla_\theta\mathcal{J}_{\text{DSM}}(x,\widetilde{x},v;\theta)$ and the condition that $\|\nabla_\theta\mathcal{J}_{\text{DSM}}(x,\widetilde{x},v;\theta)\|_2 > 0$ on the compact set. So we only need to choose $\xi$ as

$$
\xi = \frac{\sin\eta\cdot\min_{(\widetilde{x},\theta)\in B,\|v\|_2<\epsilon_0}\|\nabla_\theta\mathcal{J}_{\text{DSM}}(x,\widetilde{x},v;\theta)\|_2}{\dim(\mathcal{S})\cdot\max_{\omega\in\theta}M_\omega}. \tag{37}
$$

When we choose $\|v\|_2 = \epsilon < \min(\epsilon_0,\xi)$, we can guarantee the angle between $\nabla_\theta\mathcal{J}_{\text{FD-DSM}}(x,\widetilde{x},v;\theta)$ and $\nabla_\theta\mathcal{J}_{\text{DSM}}(x,\widetilde{x},v;\theta)$ to be uniformly less than $\eta$ on $B$. $\qquad\square$

### B.6  Application on the latent variable models

For the latent variable models (LVMs), the log-likelihood is usually intractable. Unlike EBMs, this intractability cannot be easily eliminated by taking gradients. Recently, the proposed SUMO [11] can provide an unbiased estimator for the intractable $\log p_\theta(x)$, which is defined as

$$
\text{SUMO}(x) = \text{IWAE}_1(x) + \sum_{k=1}^{K}\frac{\Delta_k(x)}{\mathbb{P}(\mathcal{K}\geq k)}, \text{ where } K\sim p_k(K) \text{ and } K\in\mathbb{N}^+. \tag{38}
$$

There are $\mathbb{P}(\mathcal{K}=K) = p_k(K)$ and $\Delta_k(x) = \text{IWAE}_{k+1}(x) - \text{IWAE}_k(x)$, where $\text{IWAE}_k(x)$ is the importance-weighted auto-encoder [2], defined as

$$
\text{IWAE}_k(x) = \log\frac{1}{k}\sum_{j=1}^{k}\frac{p_\theta(x|z_j)p_\theta(z_k)}{q_\phi(z_k|x)}, \text{ where } z_k\overset{i.i.d}{\sim}q_\phi(z|x). \tag{39}
$$

Now we can derive an upper bound for our FD reformulated objectives exploiting SUMO. To see how to achieve this, we can first derive a tractable lower bound for the first-order squared term as

$$\mathbb{E}_{p_\epsilon(v)}\mathbb{E}_{p_{\text{data}}(x)}\left[(\log p_\theta(x+v)-\log p_\theta(x-v))^2\right]$$

$$=\mathbb{E}_{p_\epsilon(v)}\mathbb{E}_{p_{\text{data}}(x)}\left[\left(\mathbb{E}_{p_k(K_1),p_k(K_2)}\left[\text{SUMO}(x+v;K_1)-\text{SUMO}(x-v;K_2)\right]\right)^2\right] \quad (40)$$

$$\leq\mathbb{E}_{p_{\text{data}}(x)}\mathbb{E}_{p_k(K_1),p_k(K_2)}\mathbb{E}_{p_\epsilon(v)}\left[\left(\text{SUMO}(x+v;K_1)-\text{SUMO}(x;K_2)+2\right)^2\right],$$

as well as a tractable unbiased estimator for the second-order term as

$$\mathbb{E}_{p_\epsilon(v)}\mathbb{E}_{p_{\text{data}}(x)}\left[\log p_\theta(x+v)+\log p_\theta(x-v)-2\log p_\theta(x)\right]$$

$$=\mathbb{E}_{p_{\text{data}}(x)}\mathbb{E}_{p_k(K_1),p_k(K_2),p_k(K_3)}\mathbb{E}_{p_\epsilon(v)}\left[\text{SUMO}(x+v;K_1)+\text{SUMO}(x-v;K_2)\right. \quad (41)$$

$$\left.-2\cdot\text{SUMO}(x;K_3)\right],$$

where we adjust the order of expectations to indicate the operation sequence in implementation. According to Eq. (40) and Eq. (41), we can construct upper bounds for our FD-SSM and FD-DSM objectives, and then train the LVMs via minimizing the induced upper bounds. In comparison, when we directly estimate the gradient-based terms $v^\top\nabla_x\log p_\theta(x)$ and $v^\top\nabla_x^2\log p_\theta(x)v$, we need to take derivatives on the SUMO estimator, which requires technical derivations [11].

### B.7 Connection to MPF

We can provide a naive FD reformulation for the SSM objective as

$$\mathcal{R}(\theta)=\frac{1}{2\epsilon^2}\mathbb{E}_{p_{\text{data}}(x)}\mathbb{E}_{p_\epsilon(v)}\left[(\log p_\theta(x+v)-\log p_\theta(x))^2+4(\log p_\theta(x+v)-\log p_\theta(x))\right]$$

$$=\frac{1}{\epsilon^2}\mathbb{E}_{p_{\text{data}}(x)}\mathbb{E}_{p_\epsilon(v)}\left[\frac{1}{2}\left(v^\top\nabla_x\log p_\theta(x)\right)^2+v^\top\nabla_x^2\log p_\theta(x)v+o(\epsilon^2)\right]=\mathcal{J}_{\text{SSM}}(\theta)+o(1). \quad (42)$$

Minimum probability flow (MPF) [14] can fit probabilistic model parameters via establishing a deterministic dynamics. For a continues state space, the MPF objective is

$$\text{K}_{\text{MPF}}=\mathbb{E}_{p_{\text{data}}(x)}\int g(y,x)\exp\left(\frac{E_\theta(x)-E_\theta(y)}{2}\right)dy,$$

where $E_\theta(x)=-\log p_\theta(x)-\log Z_\theta$ is the energy function. Let $B_\epsilon(x)=\{x+v|\|v\|_2\leq\epsilon\}$ and we choose $g(y,x)=\mathbb{1}(y\in B_\epsilon(x))$ be the indicator function, then the MPF objective becomes

$$\hat{\text{K}}_{\text{MPF}}=V_\epsilon\mathbb{E}_{p_{\text{data}}(x)}\mathbb{E}_{p_\epsilon(v)}\left[\exp\left(\frac{\log p_\theta(x+v)-\log p_\theta(x)}{2}\right)\right],$$

where $V_\epsilon$ denotes the volume of $d$-dimensional hypersphere of radius $\epsilon$. Let $\Delta_\theta(x,v)=\log p_\theta(x+v)-\log p_\theta(x)$, then we can expand the exponential function around zero as

$$\hat{\text{K}}_{\text{MPF}}=V_\epsilon\mathbb{E}_{p_{\text{data}}(x)}\mathbb{E}_{p_\epsilon(v)}\left[1+\frac{\Delta_\theta(x,v)}{2}+\frac{\Delta_\theta(x,v)^2}{8}+o(\Delta_\theta(x,v)^2)\right]=\frac{V_\epsilon\epsilon^2}{4}\left[\mathcal{R}(\theta)+o(1)\right]+V_\epsilon,$$

where the second equation holds because $\Delta_\theta(x,v)=\Theta(\epsilon)$. In this case, after removing the offset and scaling factor, the objective of MPF is directly equivalent to $\mathcal{R}(\theta)$ as to an $o(1)$ difference.

## C Implementation details

In this section, we provide a pseudo code for the implementation of FD formulation for both SSM and DSM. Then we provide the specific details in our experiments.

### C.1 Pseudo codes

The pseudo code of FD-SSM is as follows:

```
cat_input = concatenate([data, data + v, data - v], dim=0)
energy_output = energy(cat_input)
energy1, energy2, energy3 = split(energy_output, 3, dim=0)

loss1 = (energy2 - energy3)**2 / 4
loss2 = (-energy2 - energy3 + 2 * energy1) * 2
FD_SSM_loss = (loss1 + loss2).mean() / eps ** 2
```

The pseudo code of FD-DSM is as follows:

```
pdata = data + noise
cat_input = concatenate([pdata + v, pdata - v], dim=0)
energy_output = energy(cat_input)
energy1, energy2 = split(energy_output, 2, dim=0)

loss1 = (energy2 - energy1) * 0.5
loss2 = sum(v * noise/sigma, dim=-1)
FD_DSM_loss = ((loss1 + loss2)**2).mean() / eps ** 2
```

### C.2 Implementation details and definitions

**DKEF** defines an unnormalized probability in the form of $\log \tilde{p}(x) = f(x) + \log p_0(x)$, with $p_0$ is the base measure. $f(x)$ is defined as a kernel function $f(x) = \sum_{i=1}^{N} \sum_{j=1}^{N_j} k_i(x, z_j)$, where $N$ is the number of kernels, $k(\cdot, \cdot)$ is the kernel function, and $z_{j_{0 < j < N_j + 1}}$ are $N_j$ inducing points. We follow the officially released code from Song et al. [16]. Specifically, we adopt three Gaussian RBF kernel with the extracted by a three-layer fully connected neural network (NN) with 30 hidden units. The width parameters for the Gaussian kernel is jointly optimized with the parameters of the NN. We apply the standard whitening process during training following Wenliang et al. [17] and Song et al. [16]. We adopt Adam optimizer [6] with default momentum parameters and the learning rate is 0.01. The only extra hyper-parameter $\epsilon$ in the finite-difference formulation is set to 0.1.

**Deep EBM** directly defines the energy function with unnormalized models using a feed forward NN $f(\cdot)$ and the probability is defined as $p(x) = \frac{\exp(-f(x))}{\int \exp(-f(x))dx}$. The learning rate for DSM is $5 \times 10^{-5}$ and the learning rate for SSM is $1 \times 10^{-5}$ since the variance of SSM is larger than DSM. The optimizer is Adam with $\beta_1 = 0.9$ and $\beta_2 = 0.95$. The sampling method is annealed SGLD with a total of $2,700$ steps. The $\epsilon$ in the finite-difference formulation is set to 0.05. When training with annealed DSM, the noise level is an arithmetic sequence from 0.05 to 1.2 with the same number of steps as the batch size. The default batch size is 128 in all our experiments unless specified. The backbone we use is an 18-layer ResNet [4] following Li et al. [8]. No normalizing layer is used in the backbone and the output layer is of a generalized quadratic form. The activation function is ELU. All experiments adopt the ResNet with 128 filters. During testing, we randomly sample 1500 test data to evaluate the exact score matching loss.

**NICE** is a flow-based model, which converts a simple distribution $p_0$ to the data space $p$ using a invertible mapping $f$. In this case, the probability is defined as $\log p(x) = \log p_0(z) + \log \det(\frac{\partial z}{\partial x})$, where $z = f^{-1}(x)$ and $\det(\cdot)$ denotes the determinant of a matrix. The NICE model has 4 blocks with 5 fully connected layers in each block. Each layer has $1,000$ units. The activation is Softplus. Models are trained using Adam with a learning rate of $1 \times 10^{-4}$. The data is dequantized by adding a uniform noise in the range of $[-\frac{1}{512}, \frac{1}{512}]$, which is a widely adopted dequantization method for training flow models. The $\epsilon$ in the finite-difference formulation is set to 0.1.

**NCSN** models a probability density by estimating its score function, i.e., $\nabla_x \log p(x)$, which is modeled by a score net. We follow Song and Ermon [15] and provide an excerpt on the description of the model architecture design in the original paper: "We use a 4-cascaded RefineNet [10] and pre-activation residual blocks. We replace the batch normalizations with CondInstanceNorm++ [3], and replace the max-pooling layers in Refine blocks with average pooling. Besides, we also add CondInstanceNorm++ before each convolution and average pooling in the Refine blocks. All

activation functions are chosen to be ELU. We use dilated convolutions [18] to replace the subsampling layers in residual blocks, except the first one. Following the common practice, we increase the dilation by a factor of 2 when proceeding to the next cascade. For CelebA and CIFAR-10 experiments, the number of filters for layers corresponding to the first cascade is 128, while the number of filters for other cascades are doubled. For MNIST experiments, the number of filters is halved."

## C.3  Details of the results on out-of-distribution detection

For out-of-distribution (OOD) detection, we apply the typicality [12] as the detection metric. Specifically, we first use the training set $\mathcal{D}_{\text{train}}$ to approximate the entropy of model distribution as

$$\mathbb{H}[p_\theta(x)] \approx \frac{1}{N} \sum_{x \in \mathcal{D}_{\text{train}}} -\log p_\theta(x), \tag{43}$$

where $|\mathcal{D}_{\text{train}}| = N$ indicates the number of elements in the training set. Then give a set of test data $\mathcal{D}_{\text{test}}$, where we control $|\mathcal{D}_{\text{test}}| = M$ as a hyperparameter, then we can calculate the typicality as

$$\left| \left( \frac{1}{M} \sum_{x \in \mathcal{D}_{\text{test}}} -\log p_\theta(x) \right) - \mathbb{H}[p_\theta(x)] \right|. \tag{44}$$

Note that the metric in Eq. (44) naturally adapt to unnormalized models like EBMs, since there is

$$
\begin{aligned}
\frac{1}{M} \sum_{x \in \mathcal{D}_{\text{test}}} -\log p_\theta(x) &= Z_\theta + \frac{1}{M} \sum_{x \in \mathcal{D}_{\text{test}}} -\log \widetilde{p}_\theta(x); \\
\frac{1}{N} \sum_{x \in \mathcal{D}_{\text{train}}} -\log p_\theta(x) &= Z_\theta + \frac{1}{N} \sum_{x \in \mathcal{D}_{\text{train}}} -\log \widetilde{p}_\theta(x),
\end{aligned}
\tag{45}
$$

where the intractable partition function $Z_\theta$ can be eliminated after subtraction in Eq. (44). Thus we can calculate the typicality for EBMs as

$$\left| \left( \frac{1}{M} \sum_{x \in \mathcal{D}_{\text{test}}} -\log \widetilde{p}_\theta(x) \right) - \left( \frac{1}{N} \sum_{x \in \mathcal{D}_{\text{train}}} -\log \widetilde{p}_\theta(x) \right) \right|. \tag{46}$$

As shown in Nalisnick et al. [12], a higher value of $M$ usually lead to better detection performance due to more accurate statistic. Thus to have distinguishable quantitative results, we set $M = 2$ in our experiments. As to training the deep EBMs for the OOD detection, the settings we used on SVHN and CIFAR-10 are identical to those that we introduced above. On the ImageNet dataset, the images are cropped into a size of 128×128, and we change the number of filters to 64 limited by the GPU memory. On SVHN and CIFAR-10, the models are trained on two GPUs, while the model is trained on eight GPUs on ImageNet. For all datasets, we use $N = 50,000$ to estimate the data entropy and randomly sample $1000M$ test samples to conduct OOD detection.

## C.4  Results on the VAE / WAE with implicit encoders

VAE / WAE with implicit encoders enable more flexible inference models. The gradient of the intractable entropy term $H(q)$ in the ELBO can be estimated by a score net. We adopt the identical neural architectures as in Song et al. [16]. The encoder, decoder, and score net are both 3-layer MLPs with 256 hidden units on MNIST and 4-layer CNNs on CelebA. For MNIST, the optimizer is RMSProp with the learning rate as $1 \times 10^{-3}$ in all methods. The learning rate is $1 \times 10^{-4}$ on CelebA. All methods are trained for 10K iterations. The $\epsilon$ in the finite-difference formulation is set to $0.1$.