[Reviews · NeurIPS 2020]

Review 1

Summary and Contributions: This paper first details the computational costs of higher-order derivatives in neural networks through the lens of score matching (both denoising and sliced variants), whose objectives rely on higher-order expressions. The paper then develops finite difference theory to express approximations to these higher-order derivatives solely in terms of direct function evaluations, developing novel alternatives to both sliced and denoising score matching. Finally, the paper empirically validates the improved time and memory footprint of the proposed finite difference method on a variety of tasks.

Strengths: In terms of presentation, there is a clear narrative running throughout. This begins with outlining both denoising and sliced score matching, as well as the computational costs of evaluating the corresponding objectives. The paper then proceeds to address the shortcomings of the higher-order derivative expense for these score matching approaches. I don't have the numerical analyisis background to be sure, but Theorem 1 seems to introduce quite general machinery, for which only a very special case (149: K = alpha = beta = 1) is actually used. I appreciate the efficiency testing of the FD methods in Figure 3, the clear explanation of how the FD methods apply to score matching in Section 4, as well as the care taken to ensure consistency under stochastic optimization in Section 5. Finally, it's nice to see empirical evaluation across a range of tasks.

Weaknesses: My main issues concern the empirical evaluation. First, the experiments possibly seem a bit rushed, maybe because there is so much ground covered. It might generally be advisable to move some of the appendix details to the full paper. A short description of the DKEF setup, as well as the latent variable model setup, would be helpful, rather than referring the reader to either the appendix or previous work. Additionally, I'm not entirely sure of the purpose of the flow-based experiments in 6.2. Table 3 shows SM methods doing better on the SM loss, MLE doing better on NLL, and the FD-SSM and MLE times are the same, so what's the takeaway? Why would you want to fit a flow-based model using score matching where you have exact likelihood evaluation?

Correctness: I have not extensively checked the correctness of the proofs in the appendix, but the theoretical claims and arguments seem correct. The empirical evaluation is correct to the best of my knowledge, and goes toward demonstrating the promised efficiency of the proposed methods.

Clarity: The paper is generally well-written, with some caveats I've mentioned in the additional feedback section -- I've tried to mention details which really should be addressed rather than subjective stylistic points.

Relation to Prior Work: The paper clearly outlines the past work on denoising and sliced score matching, explains the expense in computing their associated objectives, and demonstrates how the proposed finite difference methods improve on these shortcomings. Again, I don't have the numerical analysis background to be able to judge the novelty of the Tth order derivative approximations, and whether similar ideas have already been explored in that literature.

Reproducibility: Yes

Additional Feedback: Concerning the title (and I know this is maybe pedantic), 'Efficient' is an entirely relative term, and means little in the absolute context of a paper title. An alternative title 'Generative Modeling via Finite-Difference Score Matching' is both clearer and more concise. What does it mean to perform learning in a generative modeling 'efficiently'? The proposed methods achieve efficiency with respect to the previous implementations of SSM and DSM, and are strictly in that sense efficient. Including 'efficient' in the title only serves to obfuscate the actual contribution of the paper, which isn't about efficient learning of generative models compared to all other generative model learning approaches, but efficient score matching by finite difference methods specifically. 20-21: '(EBMs) have better stability and mode coverage in training.' EBMs are generally considered significantly harder to train than explicitly normalized likelihood-based models, evidenced by the large literature of proposed methods for training them. In what sense can you claim EBMs are generally more stable or provide better mode coverage than e.g. a likelihood-based model? My overall issue with this sentence is that I feel it presents EBMs as more appealing than they really might be, and sweeps lots of issues under the rug. 25-26: Maybe pedantic, but the Fisher divergence doesn't directly depend on the Hessian trace, but rather can be shown to. 30: 'high computation' -> 'expensive computation' maybe? 131: 'Taylor's expansion' -> 'Taylor expansion' 134 & 140: It might be good to specify that the lemmas correspond to o() estimators for the approximation error, as opposed to any computational cost. 150: 'the generative modeling' -> 'generative modeling' 151: 'high-order' -> 'higher-order'? 152: 'optimizing on the' -> 'optimizing the' 153: 'as out future work' -> 'to future work' 193: Is this the 2-norm so that the projection is sampled uniformly on the hypersphere? 204: 'can also instantiate for' -> 'can also be used for' 208: 'element-wisely' -> 'element-wise' 231: The first clause doesn't lead to the second, maybe change to 'optimizing ... means that the model parameters ...' Table 2 caption: 'non-paralleled' -> 'non-parallelized' 251: What is a 'wide value range of epsilon'? 264: 'Except for the' -> 'In addition to' Table 3: Bolding is odd considering the significant overlap in error bars? Table 4 caption: 'in typicality' -> 'typically' ---------------------------------------------------- POST-REBUTTAL UPDATE ---------------------------------------------------- I'd like to thank the authors for their response, and would like to see the paper accepted. On learning flows by score matching: in the time since the review I came across Lyu 2009 'Interpretation and Generalization of Score Matching', which shows that the score matching objective measures how much the KL divergence between the data distribution and model would change if we perturbed both with infinitesimal Gaussian noise. Since your experiments demonstrate similar wall-clock time between FD-SSM and MLE, this correspondence could be a nice way to discuss the different properties of the learned models and why you might choose MLE or SM: MLE finds parameters which directly minimize KL, while SM finds parameters which make the KL stable to small perturbations of the data and model distribution.


Review 2

Summary and Contributions: The paper proposes estimators for higher-order directional derivatives which are more computationally efficient than reverse-mode autodiff, and applies these to training generative models with score matching (and its variants). The proposed estimators are accurate, i.e. they have error O(epsilon). They're also efficient: they require a number of function evaluations linear in the order (whereas autodiff is exponential in the order), and furthermore these function evaluations are independent so they parallelize well.

Strengths: The proposed method is sound and elegant; its advantages are clear even without considering the experiments. Sliced score matching is a relevant algorithm currently, and the finite-difference estimator makes it more efficient without any real downsides that I can see.

Weaknesses: The biggest weakness I can see is that the practical improvements are somewhat marginal, but I don't think this alone is a sufficient reason for rejection. The proposed method really shines for high-order directional derivatives, but SSM and DSM require only second- and third-order derivatives respectively. Figure 2 highlights this. As a result, the final speedup isn't dramatic (looking at the experiments, it's maybe a factor of two). I'm also not sure about the variance introduced in FD-DSM by the random projection trick (Appendix B.4); it seems to me like this might negate the speedup? A plot comparing DSM and FD-DSM in terms of loss vs. wall-clock time could demonstrate overall improvement despite the extra variance.

Correctness: I couldn't find any mistakes in the precise claims. As a disclaimer, I didn't read the Appendix proofs. Regarding the experiments, the large error intervals in Table 3 (and the lack of error intervals in the other tables) makes it hard for me to conclude which differences are significant, so the claim "our methods produce results comparable to the gradient-based counterparts" is hard to verify. I'd suggest repeating each experiment many times and reporting means and standard errors. The experiments also report training times per iteration, but these aren't directly comparable when the estimator isn't exact. A more appropriate measurement might be something like "wall-clock time to a certain loss".

Clarity: The language is clear and unambiguous. An example of the 1D case presented before the main result could make the method more accessible, but this is a minor suggestion.

Relation to Prior Work: The paper does a very good job of placing itself in the context of relevant prior work (generative modeling, score matching) and explaining the computational problem with computing higher-order derivatives using autodiff.

Reproducibility: Yes

Additional Feedback:


Review 3

Summary and Contributions: The authors propose replacing score functions (\nabla_x \log p(x)) of score-matching objectives with randomized directional derivatives -- approximated by finite difference methods -- to improve computational efficiency of training. These methods have applications in energy-based models, where score-matching is used to sidestep calculation of the partition function. Unfortunately, scores themselves are relatively expensive to calculate: hence, the paper. The authors show that these methods are faster to train while keeping equivalent performance to models trained with the original objective. This paper is well-motivated, both theoretically and empirically. The writing is clear, the results extensive, with a very references section quite broad.

Strengths: There are many strengths to this paper: - The types of finite differences used are well motivated theoretically and empirically - The number of models used to validate the method -- energy-based models, flow-based generative models, latent-variable models, and score-based generative models -- is extensive. Furthermore, the number of datasets used for each of these models (particularly the energy-based ones) is impressive. - The speedup across various models and datasets is fairly consistent. - The writing is very clear. The problem is well-posed, as is the solution.

Weaknesses: I have some pretty minor comments: - In the introduction and S2.2, the authors discuss both the time and memory issues with naive score matching. In the experimental section, only Table 2 has results on the memory saved, while the other results do not. Does the proposed method also improve memory usage on other models and datasets? - I'm pretty surprised that previous work does not exist for Lemma and Theorem 1. Although this is not my area of expertise, I would have assumed that this theory would already exist in an area such as Finite Difference Methods for Differential Equations. I know most of the teaching material focuses on the one-dimensional case, but I would be surprised if multivariate versions do not exist.

Correctness: I found the claims/method correct and the empirical methodology robust. The only minor comment I would make is that while the authors state that the FD reformulation of NCSN was only test computational efficiency, I am surprised that FID is 15 points worse than the published result.

Clarity: Yes. Other than a few typos listed below, my only suggestions would be to move S6.3 to the end of S6, since those results are in the appendix. Also, and this is very minor, sometimes a reference includes an author name, which is superfluous. Typos line 3: in respect to -> with respect to line 8: finite difference -> finite differences line 66: like the energy-based ones -> such as the energy-based ones line 81: matchs -> matches line 264: Except for the... (I'm not quite sure what you meant here) line 306: It is proved -> It is proven line 315ff: the last sentence is a bit awkward

Relation to Prior Work: Very clear. The references section is pretty extensive. My only concern is that there is previous work for this type of finite difference method, but the real novelty in the paper is that replacing SM objectives with this approximation works well.

Reproducibility: Yes

Additional Feedback: ### Update Despite some concern on the theoretic novelty in the discussion, I still think this is a very solid paper.


Review 4

Summary and Contributions: The paper proposed a new method based on finite difference (FD) to speed up the computation of any objective that involves a Hessian-vector product and specifically demonstrates how it can be applied to speed up the training of EBMs via the score matching objective. The idea is based on the observation that gradient-vector or Hessian-vector computation can be viewed as a projection operator that can be then re-written as directional derivatives, which can be well approximated via FD. The paper has established condition on which the FD has an estimation with O(1) error and showed how to construct such estimators. Empirically, the paper has shown that with the proposed FD method, two of the popular scalable estimators, SSM and DSM, can be speed up by 2.5x and 1.5 respectively in general and requires less memory.

Strengths: The paper targets an important problem of speeding up the training of EBMs via score matching, and introduces a trick which is not only theoretically and empirically useful for estimating the score matching objective, but also has its potential to use in many machine learning problems. The proposed method itself is very neat with some necessary theoretical analysis. Importantly, the consistency of FD-SSM under SGD has been established, which is necessary as the proposed method would be mostly used under a stochastic optimization setting. The empirical speed-up and memory-reduction are exciting. Although the paper focuses on applying the trick to the score matching objective, the proposed trick has a lot of potential in a wider range of applications, as the author(s) also mentioned.

Weaknesses: The method itself requires tweaks for the objectives to make it applicable, which might not be obvious in some cases. The paper jumps to "large scale" experiments directly after introducing the method. It would be ideal to have some "toy examples" which illustrate and evaluate the proposed method and contrast them against SSM and DSM. * Specifically, it is claimed that the proposed method is insensitive to a wide range of \epsilon but there is no experiment that supports this claim. * It would be nice to see the actual trace plot of the SM loss for FD and non-FD objectives, which show how good the approximations are and provide empirical evidence of the validity of Theorem 2. For the large-scale experiments: * Does FD-SSM refer to FD-SSMVR? * Why sometimes only FD-SSM is reported and sometimes only FD-DSM is report? ########## # Update # ########## The additional plot Fig. A(b) looks great to me. The y-axis of the current version is the time but I assume both are run for the same iterations. It would be good to use iterations I guess because we just want to check if FD affect the convergence rather than comparing the computational efficacy. But overall, the response looks good to me. Thanks!

Correctness: The sketch of proofs in the main paper for Theorem 1 in Section 3.2 looks correct to me. Regarding proof of Lemma 1 in the appendix: - Line 7: the notation for T + 1 real-value numbers for \gamma is weird. - \phi_t is (3) not defined. Theorem 2 looks a bit counter intuitive as the gradient estimators is biased which usually has no guarantee under stochastic optimisation. it seems that the proof is established by taking \epsilon to 0 and making the compact set B large enough to include the stationary point. Can the author(s) highlight a bit more on why it's always possible?

Clarity: The paper is very well-written and the reading of the manuscript in general is enjoyable.

Relation to Prior Work: The related work section is well-written. Through the first part of paper. how the proposed method is motivated by and built on prior works is also well-discussed.

Reproducibility: Yes

Additional Feedback: It would be nice if the author(s) can make some comments on the (empirical, performance-wise) comparison to other types of AD method (e.g. mixed-mode or source-to-source), although I agree that reverse-mode is the baseline that is needed to target primarily.

[Author Response · NeurIPS 2020]



Figure A: **(a)** Loss for DSM and FD-DSM; **(b)** Loss for SSM and FD-SSM; **(c)** FID scores on 1,000 samples (higher than those reported on 50,000 samples in the paper) for SSMVR and FD-SSMVR; **(d)** Score distribution on toy example trained by FD-SSM. The ranges of x and y axis are both $[-8, 8]$.

We thank all the reviewers for their valuable comments. Below, we address the detailed comments of each reviewer.

**To Reviewer #1. Empirical evaluation:** Thank you for the suggestion. In the revision, we will move more details into
the full paper to improve the reading experience. **Flow-based models:** For fair comparisons with the SM baselines
(e.g., SSM), our evaluation tasks mainly follow the settings in Song et al. [57], including the flow-based experiments.
Technically, Table 3 shows that our method can also outperform on invertible architectures, where we can regard
flow-based models as special instantiations of EBMs. **About additional feedback:** We really appreciate for the detailed
caveats list, and we will carefully polish them in the final version. As to the reasonable suggestion on the title, we will
also discuss on it, thank you! Below we answer the remaining questions in the list. 20-21: Here the better stability
and mode coverage are compared to GANs. We will make the claim more precise. 193: Yes, the subscript of norm
should be 2, which is a uniform distribution on the hypersphere. 251: For deep EBMs, we experiment on the range of
$\epsilon \in [0.01, 1]$ and obtain comparable model performance. We will include complete results in the revision.

**To Reviewer #2. Practical improvements:** Indeed, SM methods only involve second- or third-order derivatives, which
may not result in dramatic speedup using our FD reformulation. Thus our main future work is to better exploit our
FD formulas in other higher-order cases. Some potential candidates include applications in meta-learning and those
described in L150-153. **Variance of FD:** As discussed in L314-317, the random projection trick required by the FD
formula is its main downside, which may outweigh the gain on efficiency for low-order computations. We provide
the loss curve of DSM and FD-DSM w.r.t. time in Fig. A(a). As seen, FD-DSM can achieve the best model (lowest
SM loss) faster, but eventually converges to higher loss compared to DSM. In contrast, as shown in Fig. A(b)(c), when
applying FD on SSM-based methods, the improvements are much more significant. **Repeating experiments:** Thank
you for the suggestion. We will repeat the experimental trials more times to alleviate the effect of randomness.

**To Reviewer #3. Memory usage:** Our method also consistently improve the memory usage across different models
and datasets. For example, on the NCSN model, SSM and FD-SSM use 6.3G and 5.4G memory per GPU (both run on
four GPUs), respectively. We will include more complete results on memory usage in the revision. **Previous work
on Lemma and Theorem 1:** As far as we know, we can only find out the reference like [21] on the univariate case
(still slightly different from the one-dimensional instantiation of our formulas). We will keep searching for the related
literature on the multivariate versions and add proper references then. **Performance gap of NCSN:** The published FID
results in NCSN are based on DSM. Although DSM is a biased estimator, it naturally adapts to the annealed Langevin
dynamics used in NCSN, compared to SSM. As in L284-285, our NCSN experiment is mainly to demonstrate that our
method makes the unbiased SSM (VR) estimator more efficient on score-based networks. We will explain this clearly.
**Clarity:** We really appreciate the suggestions and the typos list. We will carefully polish them in the revision.

**To Reviewer #4. Applicable cases:** Our FD formula is a general technique to estimate the directional derivatives and
their related objectives. Some typical applicable cases include gradient norms, gradient projection, Hessian trace, and
Hessian Frobenius norm, etc. **Toy example:** We adopt the data distribution $0.8\mathcal{N}([5,5],I)+0.2\mathcal{N}([-5,-5],I)$ as a 2D
toy example. We use a three-layer MLP as our EBM model and set $\epsilon = 0.1$. The result of the score distribution trained
by FD-SSM is given in Fig. A(d), which is almost the same as SSM (full details will be added). **Sensitivity analysis
of $\epsilon$:** Here we report the test NLLs for DKEF model on the Parkinson dataset: $14.17(\epsilon = 0.1), 13.51(\epsilon = 0.05)$,
$14.03(\epsilon = 0.02), 14.00(\epsilon = 0.01)$, which indicates the insensitivity for $\epsilon \in [0.01, 0.1]$. Complete results on EBMs w.r.t.
$\epsilon$ will be involved in the revision. **Trace plot of SM loss:** The plot is given in Fig. A(b) for SSM and FD-SSM. We can
see that FD-SSM has significantly better efficiency and is consistent with SSM. **Large-scale experiments:** FD-SSMVR
is the FD version of SSMVR, as formulated in L204-212. As described in [57], SSMVR is a variance reduction form of
SSM when the projection distribution $p(v)$ satisfies certain conditions. Since we experiment on various combinations of
datasets, models, tasks, and SM methods, we attempt to make the results as diverse as possible in the limited space. We
will involve more complete results in the revision. **Proof of Theorem 2:** Our FD reformulations are asymptotically
unbiased when $\epsilon \to 0$. Thus, under the condition in Lemma 2, $\forall \delta > 0$, there $\exists \epsilon$, such that the remainder term (estimation
bias) can be uniformly bounded by $\delta$ in $B$, i.e., the gradients of FD-SSM and SSM can be sufficiently aligned in $B$.
As $\epsilon \to 0$, the set $B$ could tend to $\mathbb{R}^d \times \mathcal{S} \setminus \{\text{stationary points}\}$, and the optimization track can converge in $B$ to the
stationary points (may not exactly locate on). **Comparisons to other AD methods:** Thank you for the suggestion. We
will try to add extensive experiments on other types of automatic differentiation methods in the revision.

[Meta-Review · NeurIPS 2020]

The authors reformulate denoising score matching and sliced score matching in terms of directional derivatives (1st and 2nd order respectively) and show that estimating these derivatives using finite differences (FD) leads to faster training and lower memory usage while yielding comparable results to using exact derivatives. The paper also introduces a general approach to estimating directional derivatives of any order using FDs of the function. The reviewers found the paper interesting and very well written. They also appreciated the extensive evaluation of the algorithms on a range of different generative models as well as the clarifications provided in the rebuttal. There were some concerns however about the novelty as well as the necessity of the developed FD estimation approach for directional derivatives. The authors are encouraged to perform a more extensive literature search as well as explain why applying the standard FD machinery to the definition d/dvL(x) = lim_h->0 (L(x+hv)-L(x-hv))/(2h) (and its higher order generalizations) is insufficient to achieve the aims of the paper.